# UCPO: Uncertainty-Aware Policy Optimization

**Xianzhou Zeng** [1]  **Jing Huang** [2]  **Chunmei Xie** [1]  **Gongrui Nan** [1]  **Siye Chen** [1]  **Mengyu Lu** [1]  **Weiqi Xiong** [1]
**Qixuan Zhou** [1]  **Junhao Zhang** [2]  **Qiang Zhu** [2]  **Yadong Li** [1]  **Xingzhong Xu** [1]

## Abstract

The key to building trustworthy large language models (LLMs) lies in endowing them with inherent uncertainty expression capabilities, thereby mitigating overconfident errors in high-stakes applications. However, existing RL paradigms such as GRPO often suffer from Advantage Bias due to binary decision spaces and static uncertainty rewards, inducing either excessive conservatism or overconfidence. To tackle this challenge, this paper unveils the root causes of reward hacking and overconfidence in current RL paradigms incorporating uncertainty-based rewards, based on which we propose the **Un**Certainty-Aware **P**olicy **O**ptimization (**UCPO**) framework. UCPO employs Ternary Advantage Decoupling to separate and independently normalize deterministic and uncertain rollouts, thereby eliminating advantage bias. Furthermore, a Dynamic Uncertainty Reward Adjustment mechanism adapts uncertainty weights in real-time according to model evolution and instance difficulty. Experimental results in mathematical reasoning and general tasks demonstrate that UCPO effectively resolves the reward imbalance, significantly improving the reliability of the model beyond their knowledge boundaries. The code is available at `https://github.com/xzhouzeng/ucpo`.

## 1. Introduction

While Large Language Models (LLMs) excel in complex reasoning, their tendency toward overconfidence under uncertainty, which can lead to erroneous assertions, remains a critical barrier to high-stakes deployment (Huang et al., 2023; Cossio, 2025). Building trustworthy AI necessitates endowing models with uncertainty awareness, enabling

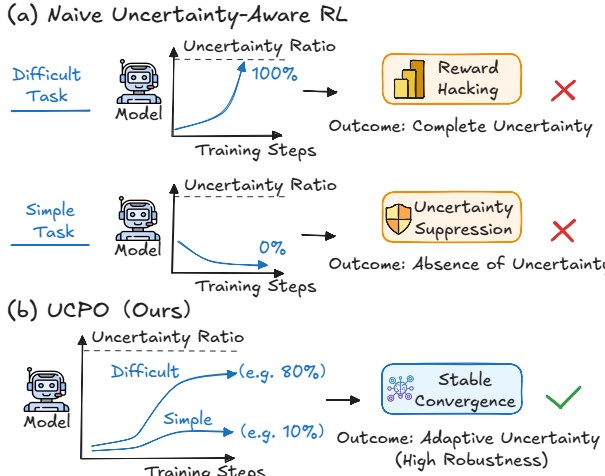

*Figure 1.* Illustration of reward imbalance in uncertainty alignment: static rewards trigger overconfidence or avoidance degeneracy, whereas UCPO stabilizes the policy through adaptive reward.

them to recognize cognitive boundaries and express doubt when queries exceed their internal knowledge (Tonmoy et al., 2024).

To equip models with uncertainty awareness, existing research primarily follows two trajectories (Wen et al., 2025). The first employs Supervised Fine-Tuning (SFT) using instructional datasets with explicit abstention labels to facilitate imitation learning (Amayuelas et al., 2024; Kapoor et al., 2024). However, the high cost of data synthesis and the inability of static datasets to capture dynamic inference-time uncertainty limit its scalability. The second approach utilizes Reinforcement Learning (RL) by assigning fixed intermediate rewards (e.g., 0.5) to uncertain responses. While this method requires no additional annotation, its efficacy is highly sensitive to reward tuning (Wei et al., 2025; Ren et al., 2025). As shown in Figure 1-(a), this naive uncertainty-aware RL paradigm lacks robustness: static rewards cannot adapt to evolving model capabilities or varying task difficulties. In difficult tasks, models often succumb to reward hacking, resulting in avoidance degeneracy where risk is mitigated through excessive refusal. Conversely, in simpler tasks, subtle uncertainty signals are frequently overwhelmed by the high rewards of correct answers, resulting in overcon-

[1]Ant Group [2]Zhejiang University. Correspondence to: Yadong Li <liyadong.lyd@antgroup.com>.

*Proceedings of the 43rd International Conference on Machine Learning*, Seoul, South Korea. PMLR 306, 2026. Copyright 2026 by the author(s).

fidence and the suppression of uncertainty cognition.

Addressing these challenges, this paper provides an in-depth analysis of the underlying mechanisms by which existing RL methods trigger reward hacking and overconfidence. We reveal that a fixed uncertainty reward mechanism leads to a bias in the Advantage Function across different performance intervals: in high-performance regimes, models fail to learn uncertainty as the uncertainty advantage turns negative; in low-performance regimes, models lapse into reward hacking due to an overloaded uncertainty advantage. Building on these insights, we propose the Uncertainty-Aware Policy Optimization (UCPO) framework, which transcends the limitations of binary decision spaces to achieve a dynamic equilibrium among truthful, erroneous, and abstinent ternary responses. Specifically, UCPO comprises two synergistic components: Ternary Advantage Decoupling (TAD), which decomposes sampling paths into independent deterministic and uncertain channels and implements Independent Advantage Normalization to eliminate mutual interference between semantic signals; and Dynamic Uncertainty Reward Adjustment (DURA), which realigns reward weights in real-time based on the evolution of model capability and sample difficulty distribution, effectively mitigating policy drift caused by static reward imbalance.

Experimental results on Qwen3 (Yang et al., 2025) and Llama3.1 (Dubey et al., 2024) show that UCPO achieves stable convergence across varying task difficulties, as illustrated in Figure 1-(b). Evaluations in mathematical reasoning and general domains confirm that UCPO effectively reduces overconfident erroneous assertions while enhancing the reliability of answered responses and explicit abstention calibration at the model's cognitive boundaries.

The contributions of this paper are summarized as follows:

- We analyze the mechanisms of advantage bias and static reward failure within uncertainty-aware reinforcement learning, identifying the root causes of overconfidence and avoidance degeneracy.

- By integrating TAD and DURA, UCPO establishes an adaptive RL paradigm for the trustworthy alignment of LLMs that eliminates the need for exhaustive reward hyperparameter tuning.

- Extensive experiments demonstrate UCPO's efficacy in mitigating overconfident errors and enhancing uncertainty expression across various tasks.

## 2. Preliminary

In this section, we first review the Group Relative Policy Optimization (GRPO) framework (Shao et al., 2024). Building on this, we delineate the Ternary Imbalance Problem

inherent in the integration of uncertainty rewards within this framework.

### 2.1. Group Relative Policy Optimization

GRPO streamlines the reinforcement learning process by obviating the need for a separate value function. Instead of estimating state values, it leverages the collective rewards of multiple outputs generated from the same prompt to derive a baseline for advantage estimation.

Specifically, for a given prompt $q$, the policy model $\pi_\theta$ generates a group of $G$ outputs $\{o_1, o_2, \ldots, o_G\}$. In a standard binary setting, each rollout $o_i$ receives a reward $r_i \in \{r_{\text{wrong}}, r_{\text{right}}\}$. The advantage $\hat{A}_{i,t}$ is computed via group-relative normalization:

$$\hat{A}_{i,t} = \frac{r_i - \text{mean}(\mathbf{r})}{\text{std}(\mathbf{r})} \tag{1}$$

where $\mathbf{r} = [r_1, r_2, \ldots, r_G]$. This ensures $\sum_{i=1}^{G} \hat{A}_{i,t} = 0$, creating a zero-sum gradient signal. Building upon this advantage estimate, the policy is updated by maximizing the GRPO objective function $\mathcal{J}_{\text{GRPO}}(\theta)$:

$$\mathcal{J}_{\text{GRPO}}(\theta) = \mathbb{E}\left[ \frac{1}{G} \sum_{i=1}^{G} \frac{1}{|o_i|} \sum_{t=1}^{|o_i|} \left( \min\left( r_{i,t}(\theta)\hat{A}_{i,t}, \right.\right.\right.$$
$$\left.\left.\left. \text{clip}(r_{i,t}(\theta), 1-\varepsilon, 1+\varepsilon)\hat{A}_{i,t} \right) - \beta D_{\text{KL}}(\pi_\theta \| \pi_{\text{ref}}) \right) \right] \tag{2}$$

where $r_{i,t}(\theta) = \pi_\theta(o_{i,t}|q, o_{i,<t})/\pi_{\theta_{\text{old}}}(o_{i,t}|q, o_{i,<t})$ is the probability ratio between the current and old policies.

### 2.2. The Ternary Imbalance Problem

To equip the model with the ability to express uncertainty, an intuitive extension is to introduce an uncertain category into the original reward space. This category is assigned a median reward value $r_{\text{uncertain}}$ (hereafter $r_{\text{u}}$), such that $r_{\text{wrong}} < r_{\text{u}} < r_{\text{right}}$ (set as 0, 0.8, and 1 respectively in Fig. 2). We refer to this direct integration of uncertainty rewards within the GRPO framework as GRPO-UC.

The transition from a binary to a ternary reward modeling introduces a fundamental optimization bias. We visualize this via ternary plots in Figure 2, analyzing two key metrics of GRPO-UC across diverse sample distributions: the normalized advantage of uncertain rollouts and the Net Right Advantage (defined as the difference between the aggregated advantages of right rollouts and uncertain rollouts). Our analysis reveals that the advantage signal for uncertain rollouts is highly sensitive to the model's evolving capability, triggering two distinct failure modes:

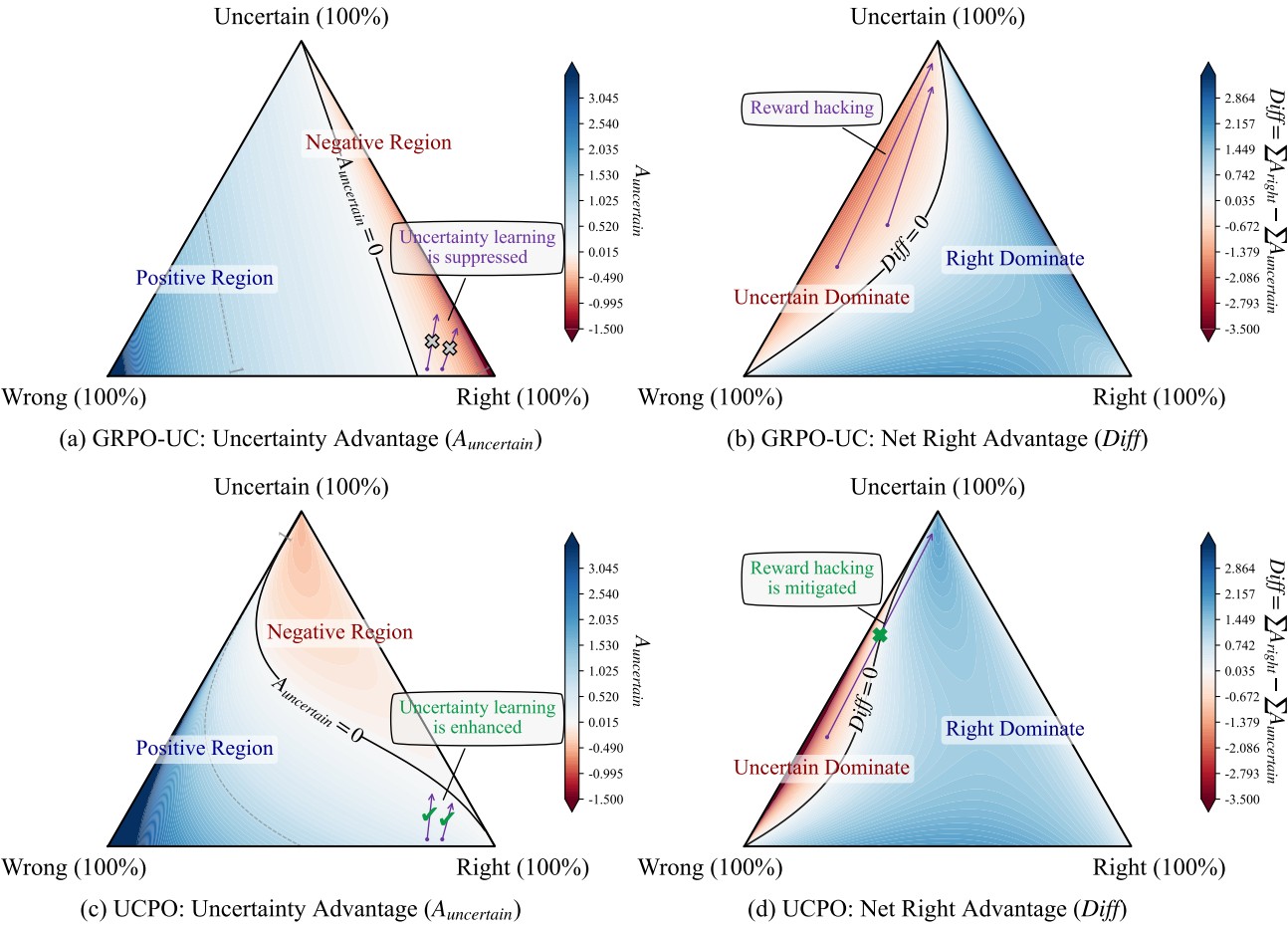

*Figure 2.* The Ternary Imbalance Problem in GRPO-UC (a-b) contrasted with the balanced advantage distribution in UCPO (c-d). Each point in the ternary plots represents a specific combination of Right, Wrong, and Uncertain proportions within a group of $G$ outputs.

- **Uncertainty Suppression in High-Performance Regimes:** As the model gains proficiency, the advantage of uncertain rollouts ($A_{\text{uncertain}}$) turns negative (red region, Fig 2.a). This illustrates Majoritarian Suppression, where the global average performance penalizes locally rational, conservative decisions. Rather than being punished for errors, the model is penalized because its cautious choice yields returns below the group's high expectations, compelling overconfidence on ambiguous samples.

- **Reward Hacking in Low-Performance Regimes:** In low-performance regimes or on high-difficulty tasks, the total advantage of uncertain rollouts dominates the policy gradient (red region, Fig 2.b). This triggers Reward Hacking, where the model identifies claiming uncertainty as a shortcut to maximize rewards without performing complex reasoning. Such behavior leads to Mode Collapse, where the model defaults to uncertainty for all inputs, stifling the incentive to learn discriminative features required for correct predictions.

The proposed UCPO breaks this ternary imbalance by restructuring the uncertain reward mechanism (Fig 2.c, d). By decoupling uncertainty rewards from global performance averages, UCPO maintains a positive advantage for uncertain rollouts in high-performance regimes where hallucinations persist, effectively preventing majoritarian suppression. Simultaneously, in low-performance regimes, it balances the gradient flow so that the uncertainty advantage does not dominate the optimization path toward an all-uncertainty policy, avoiding reward hacking and maintaining optimization pressure for learning discriminative features.

The complete mathematical derivation and theoretical justification for the ternary regions visualized in Figure 2 are provided in Appendix A.

## 3. Method

To overcome the limitations of static uncertain reward modeling, we propose Uncertainty-Aware Policy Optimization (UCPO). UCPO replaces traditional binary feedback with a

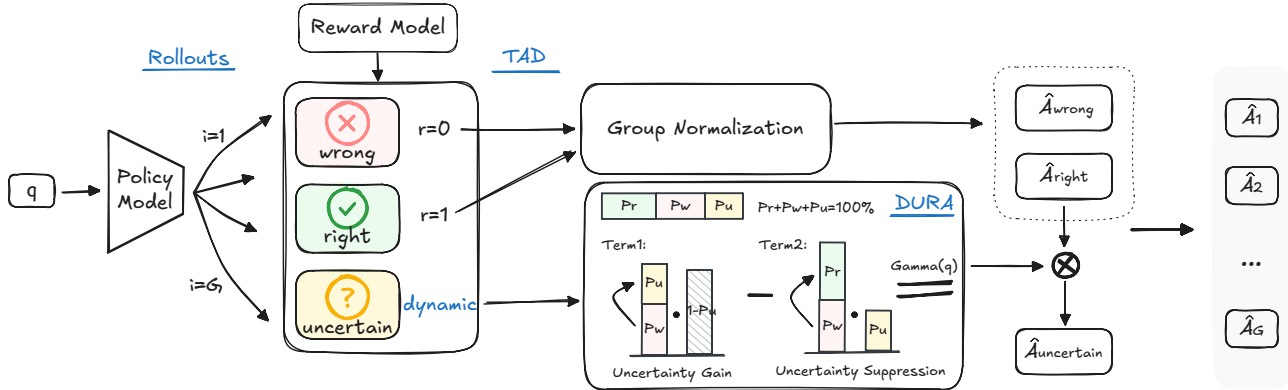

*Figure 3.* Architecture of the UCPO Framework.

ternary reward modeling system that distinguishes among right, wrong, and uncertain responses. Specifically, UCPO redefines the advantage estimation $\hat{A}_{i,t}$ through two synergetic mechanisms, as illustrated in Figure 3.

### 3.1. Ternary Advantage Decoupling (TAD)

To prevent uncertainty signals from being overshadowed by high-reward right answers, UCPO employs TAD to isolate deterministic and uncertain signals, thereby eliminating semantic interference during advantage estimation. We partition the group of $G$ rollouts into a deterministic set $\mathcal{S}_{det} = \{o \in \text{Right} \cup \text{Wrong}\}$ and an uncertainty set $\mathcal{S}_{unc} = \{o \in \text{Uncertain}\}$.

The advantage $\hat{A}_{i,t}$ is computed through two independent channels:

- **Deterministic Channel** ($o_i \in \mathcal{S}_{det}$)**:** The advantage is calculated strictly within the deterministic subset to maintain the gradient of correctness:

$$\hat{A}_{i,t}^{det} = \frac{r_i - \text{mean}(\mathbf{r}_{det})}{\text{std}(\mathbf{r}_{det}) + \epsilon} \quad (3)$$

Here, $\mathbf{r}_{det} = \{r_j \mid o_j \in \text{Right} \cup \text{Wrong}\}$ isolates core knowledge acquisition from metacognitive learning. In this channel, correct paths yield positive reinforcement ($\hat{A}_{right} > 0$), while erroneous paths provide corrective penalties ($\hat{A}_{wrong} < 0$) to suppress erroneous definitive answers.

- **Uncertainty Channel** ($o_i \in \mathcal{S}_{unc}$)**:** The advantage for expressing uncertainty is defined as a dynamic projection of the right-sample advantage:

$$\hat{A}_{i,t}^{unc} = \gamma(q) \cdot \hat{A}_{right} \quad (4)$$

By adopting $\hat{A}_{right}$ as a performance anchor, the incentive for uncertainty is dynamically scaled relative to the model's current peak reasoning capability. This anchoring prevents the signal suppression inherent in global normalization, where high right-sample density often forces uncertainty advantages into negative values, inadvertently penalizing honest doubt. By projecting $\hat{A}_{right}$ through the gain $\gamma(q)$, UCPO maintains a positive gradient to curb overconfident errors during high-error phases. The gain coefficient $\gamma(q)$, generated by the DURA module, adaptively modulates this signal.

Note that if $\mathcal{S}_{det}$ lacks either correct or incorrect rollouts, we filter these samples, a process defined as Non-Ternary Filtering (NTF). Analogous to the zero-advantage setting in standard GRPO for all-correct or all-incorrect groups, NTF effectively discards such samples to maintain training stability during extreme performance phases.

By decoupling these channels, TAD treats the expression of uncertainty as a distinct, legitimate cognitive state, preventing the model from treating it as a shortcut to bypass difficult reasoning.

### 3.2. Dynamic Uncertainty Reward Adjustment (DURA)

To maintain a stable ternary equilibrium, we introduce DURA to adaptively modulate the gain coefficient $\gamma(q)$. DURA monitors the model's real-time error rates and confidence levels to prevent both overconfidence and avoidance degeneracy through a dual-term formulation:

$$\gamma(q) = \underbrace{\left(\frac{P_w}{P_u + P_w + \epsilon}\right)(1 - P_u)}_{\text{Term 1: Uncertainty Gain}}$$
$$- w \cdot \underbrace{\left(\frac{P_r}{P_r + P_w + \epsilon}\right)P_u}_{\text{Term 2: Uncertainty Suppression}} \quad (5)$$

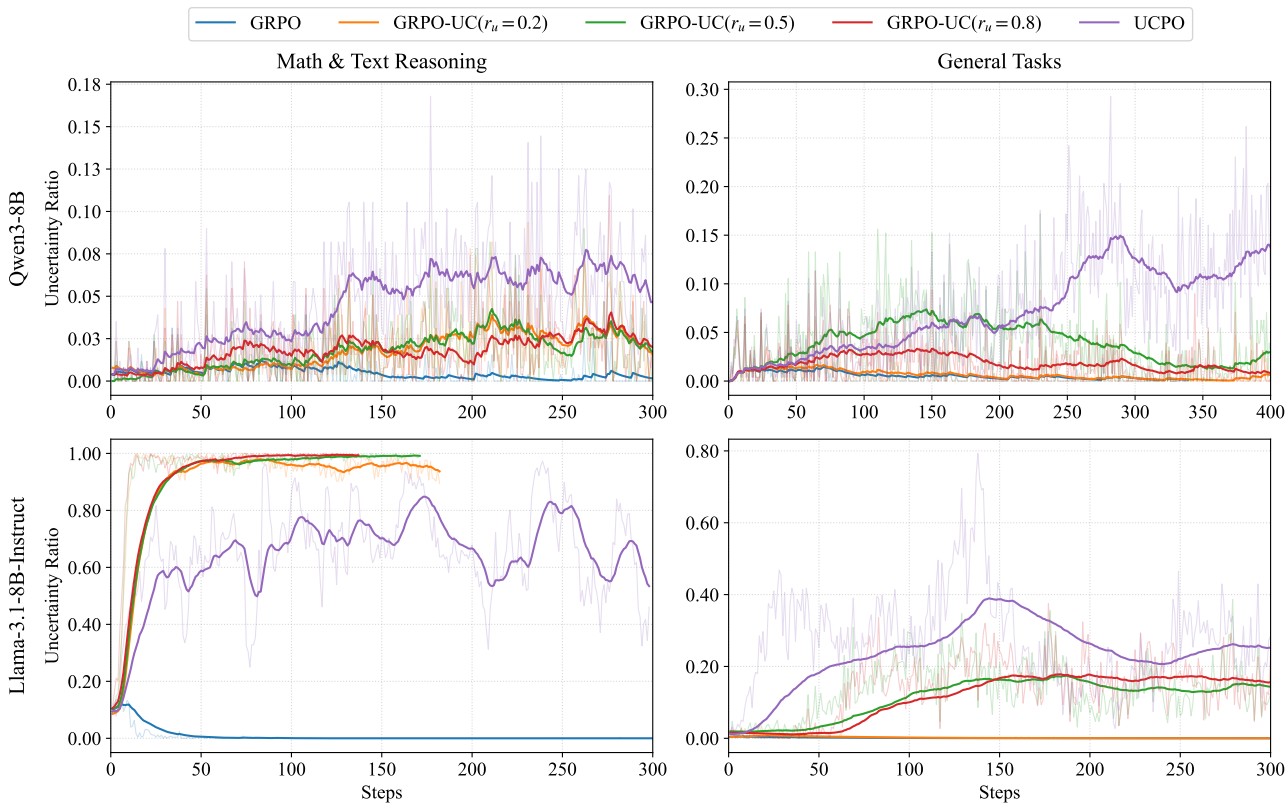

*Figure 4.* Evolution of the uncertainty ratio over training steps, comparing baseline GRPO, the proposed UCPO, and GRPO-UC variants with different reward coefficients $r_u$.

where $P_r, P_w, P_u$ denote the ratios of Right, Wrong, and Uncertain rollouts within a group, respectively, and $w = 1$ is a weighting constant. The mechanism functions as follows:

- **Uncertainty Gain (Term 1):** This term amplifies the incentive for uncertainty when the model's error-to-uncertainty ratio is high. By scaling with $(1 - P_u)$, it encourages the transition from overconfident erroneous assertions to honest doubt while preventing the policy from saturating in total avoidance.

- **Uncertainty Suppression (Term 2):** As the model's proficiency improves ($P_r$ increases), this term penalizes unnecessary avoidance. It effectively raises the competitive bar for choosing the uncertainty path, pushing the model to commit to a definitive correct answer when it has the capacity to do so, rather than defaulting to a safe but uninformative response.

DURA dynamically balances these terms, making the uncertainty channel a regulated buffer: it suppresses hallucinations early in training while driving the model toward deterministic accuracy as reasoning improves. Notably, in low-resource scenarios with limited rollouts, the gain estimation suffers from high variance and reduced dynamic range.

To address this, we propose Extensions for Low-Resource Scenarios, including batch-level smoothing and non-linear mapping; see the supplementary material for details.

## 4. Experiments

### 4.1. Experimental Settings

**Datasets and evaluation metrics.** This study evaluates cognitive boundaries and overconfidence mitigation through two task paradigms. The first, free-form Math & Text Reasoning, employs DAPO-Math-17k (Yu et al., 2025) for training and assesses performance on AIME24 (AIM, 2024), AMC (LI et al., 2024), MATH500 (Lightman et al., 2023), Minerva (Lewkowycz et al., 2022), and Olympiad Bench (He et al., 2024). The second, constrained-choice General Tasks, uses MMLU-Redux2 (Gema et al., 2025) (with 1,000 instances partitioned for testing and the remainder for training) and GPQA-Diamond (Rein et al., 2024) to scrutinize overconfidence under compelled-choice scenarios. Model responses are categorized as Accuracy ($Acc$), Hallucination ($Hal$, here referring to erroneous definitive answers), or explicit Uncertainty ($Unc$). Following KnowRL (Ren et al., 2025), we employ two primary metrics: Precision on Answered Questions (PAQ), defined as $Acc/(Acc + Hal)$, to

*Table 1.* Performance on Math & Text Reasoning tasks. † denotes the GRPO-UC reward coefficients $r_u \in \{0.2, 0.5, 0.8\}$.

| Methods | AIME24 | | AMC | | MATH500 | | Minerva | | Olympiad Bench | | Average | |
|---|---|---|---|---|---|---|---|---|---|---|---|---|
| | PAQ | F1 | PAQ | F1 | PAQ | F1 | PAQ | F1 | PAQ | F1 | PAQ | F1 |
| *Qwen3-8B* | | | | | | | | | | | | |
| Baseline | 73.33 | 73.33 | 91.57 | 91.57 | 96.80 | **96.80** | 45.96 | 45.96 | 69.63 | **69.63** | 75.46 | 75.46 |
| Prompt-UC | 73.33 | 73.33 | 89.02 | 88.48 | 95.80 | 95.80 | **49.43** | **48.60** | 71.49 | 70.47 | 75.82 | 75.34 |
| GRPO | 77.01 | 75.71 | 88.35 | 88.35 | 96.46 | 96.36 | 47.18 | 46.62 | 69.22 | 68.25 | 75.64 | 75.06 |
| GRPO-UC$^{†0.2}$ | 83.75 | **78.82** | 88.98 | 88.26 | 96.31 | 95.99 | 48.60 | 47.62 | 70.68 | 68.35 | 77.66 | 75.81 |
| GRPO-UC$^{†0.5}$ | 80.00 | 75.29 | 90.00 | 88.34 | 96.24 | 95.95 | 46.32 | 45.51 | 70.84 | 68.64 | 76.68 | 74.75 |
| GRPO-UC$^{†0.8}$ | 72.41 | 71.19 | **95.06** | **93.90** | 96.20 | 96.20 | 47.55 | 46.93 | 70.68 | 69.24 | 76.38 | 75.49 |
| UCPO | **86.11** | 76.54 | 91.95 | 89.48 | **97.28** | 96.40 | 49.15 | 47.63 | **73.67** | 69.42 | **79.63** | **75.90** |
| *Llama-3.1-8B-Instruct* | | | | | | | | | | | | |
| Baseline | 3.33 | 3.33 | 15.66 | 15.66 | 45.80 | 45.80 | 15.81 | 15.81 | 14.96 | 14.96 | 19.11 | 19.11 |
| Prompt-UC | **6.98** | **6.82** | 19.74 | 19.09 | 47.32 | 46.34 | 16.54 | 16.04 | 17.45 | 16.61 | 21.61 | 20.98 |
| GRPO | 3.33 | 3.33 | 20.08 | 20.08 | 43.96 | 43.95 | 18.16 | 18.15 | 14.76 | 14.74 | 20.06 | 20.05 |
| GRPO-UC$^{†0.2}$ | 3.85 | 2.82 | 11.80 | 9.27 | 40.98 | 33.60 | 19.88 | 11.50 | 13.58 | 9.38 | 18.02 | 13.31 |
| GRPO-UC$^{†0.5}$ | 0.00 | 0.00 | 21.43 | 6.19 | 57.61 | 25.53 | **26.16** | 9.11 | 19.28 | 4.22 | 24.90 | 9.01 |
| GRPO-UC$^{†0.8}$ | 0.00 | 0.00 | 15.79 | 2.24 | 56.61 | 12.67 | 25.00 | 4.87 | 12.90 | 1.49 | 22.06 | 4.25 |
| UCPO | 5.13 | 3.10 | **28.12** | **22.00** | **60.95** | **51.95** | 22.50 | **18.48** | **25.56** | **17.69** | **28.45** | **22.65** |

*Table 2.* Performance on General Tasks. † denotes the GRPO-UC reward coefficients $r_u \in \{0.2, 0.5, 0.8\}$.

| Methods | GPQA-Diamond | | MMLU-Redux2 | | Average | |
|---|---|---|---|---|---|---|
| | PAQ | F1 | PAQ | F1 | PAQ | F1 |
| *Qwen3-8B* | | | | | | |
| Baseline | 56.57 | 56.57 | 87.20 | 87.20 | 71.88 | 71.88 |
| Prompt-UC | 56.84 | 55.67 | 87.08 | 86.33 | 71.96 | 71.00 |
| GRPO | 59.35 | 58.79 | 87.62 | 87.21 | 73.48 | 73.00 |
| GRPO-UC$^{†0.2}$ | 60.25 | 59.06 | 88.13 | **87.61** | 74.19 | 73.33 |
| GRPO-UC$^{†0.5}$ | 65.09 | 58.65 | 88.82 | 86.61 | 76.96 | 72.63 |
| GRPO-UC$^{†0.8}$ | 64.00 | **60.05** | 88.51 | 87.51 | 76.26 | **73.78** |
| UCPO | **67.70** | 56.16 | **91.67** | 85.42 | **79.68** | 70.79 |
| *Llama-3.1-8B-Instruct* | | | | | | |
| Baseline | 22.56 | 22.56 | 65.17 | 65.17 | 43.86 | 43.86 |
| Prompt-UC | 23.28 | 22.74 | 68.02 | 67.41 | 45.65 | 45.07 |
| GRPO | 27.27 | 27.27 | 71.23 | 71.23 | 49.25 | 49.25 |
| GRPO-UC$^{†0.2}$ | 29.46 | **29.46** | 72.47 | **72.47** | 50.96 | **50.96** |
| GRPO-UC$^{†0.5}$ | 34.21 | 18.98 | 78.89 | 71.51 | 56.55 | 45.25 |
| GRPO-UC$^{†0.8}$ | 34.32 | 19.52 | 76.99 | 69.78 | 55.66 | 44.65 |
| UCPO | **36.02** | 17.18 | **81.13** | 69.03 | **58.58** | 43.10 |

measure the factual reliability of non-uncertain outputs; and the F1 Score, which balances truthfulness and informativeness by penalizing both erroneous definitive answers and excessive conservatism.

**Models and Baselines.** To assess generalizability, Qwen3-8B and Llama-3.1-8B-Instruct are selected as backbone models. Comparative baselines include the original Baseline, prompt-based uncertainty-aware guidance (Prompt-UC), and standard GRPO utilizing binary rewards ($Right = $

$1, Wrong = 0$). Additionally, GRPO-UC is included as a representative uncertainty learning strategy, employing fixed uncertainty rewards ($r_u \in \{0.2, 0.5, 0.8\}$) to highlight advantage bias and training instability in ternary decision spaces.[1]

**Training and Evaluation Details.** All experiments are conducted on a cluster of 8 A100 GPUs with a sampling group size $G = 8$ for reinforcement learning. During the evaluation phase, the decoding temperature is set to $0.6$. To ensure statistical reliability, all metrics are reported as the average performance across three independent responses generated for each instance.

### 4.2. Analysis of Training Dynamics

To investigate the underlying mechanisms of uncertainty learning, we analyze the evolution of the uncertainty ratio across diverse models and tasks. As illustrated in Figure 4, standard GRPO consistently maintains a near-zero uncertainty ratio, as its binary reward structure ($Right/Wrong$) provides no incentive for expressing doubt, resulting in persistent overconfidence. While the GRPO-UC variant attempts to address this with a fixed uncertainty reward ($r_u$), it proves highly brittle across varying task difficulties. On high-accuracy tasks (e.g., Qwen3-8B on Math &

---

[1]GRPO-UC is not intended as the name of an existing method. It serves as a controlled abstraction of the fixed-reward core shared by TruthRL/KnowRL-style uncertainty learning while excluding extra factors such as data construction and pipeline-specific training recipes, which would confound the comparison of reward mechanisms.

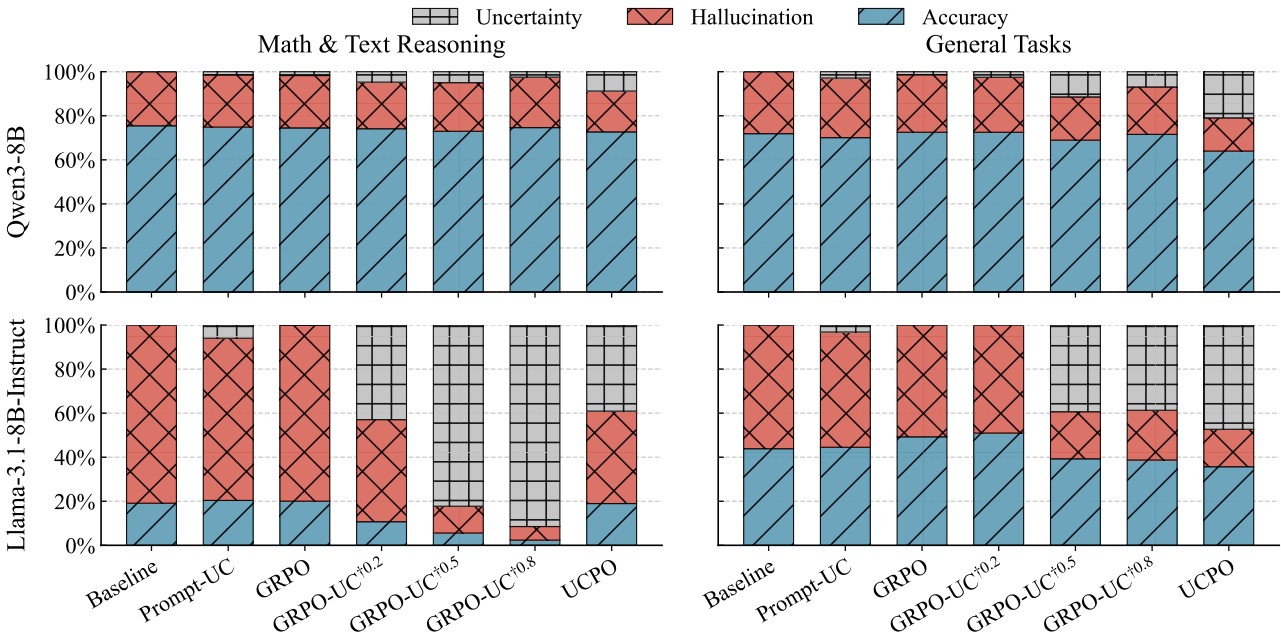

*Figure 5.* Aggregated distribution of Accuracy, Hallucination and Uncertainty across different alignment methods. Proportions are averaged over all datasets independently within the Math & Text Reasoning and General Tasks domains.

Text Reasoning), the fixed reward is insufficient to overcome the model's inherent bias toward assertion, causing the uncertainty ratio to fluctuate near 0%. Conversely, on low-accuracy tasks (e.g., Llama-3.1-8B-Instruct on Math & Text Reasoning), a high fixed reward ($r_u \geq 0.5$) triggers reward hacking, where the model prematurely abandons the exploration of correct reasoning paths in favor of guaranteed uncertainty rewards. This leads to a catastrophic collapse, with the uncertainty ratio surging to 100%.

These results indicate that UCPO effectively prevents avoidance degeneracy in complex tasks while eliciting honest expressions in simpler ones. By converting overconfident erroneous answers into explicit uncertainty expressions without sacrificing proactive problem-solving, UCPO maintains an effective trade-off between truthfulness and informativeness across varying task difficulties and model capacities. Detailed analyses of PAQ and F1 score evolution relative to training steps and uncertainty ratios are provided in the Supplementary Material.

### 4.3. Main Results

Table 1 and Table 2 present the performance across math-reasoning and general knowledge tasks, while Figure 5 visualizes the corresponding response category distributions averaged independently within each task domain. Several key findings emerge:

**Reliability of Factual Commitments.** UCPO consistently achieves the highest Average PAQ across all evaluated domains. Specifically, in Math & Text Reasoning tasks, UCPO reaches a PAQ of 79.63% on Qwen3-8B and 28.45% on Llama-3.1-8B-Instruct. In General Tasks, it similarly leads with 79.68% and 58.58% for the respective models. As illustrated in Figure 5, this performance gain is primarily driven by the strategic conversion of erroneous definitive answers (red) into honest uncertainty expressions (gray). Unlike standard GRPO, which lacks a mechanism to penalize overconfident erroneous assertions, UCPO ensures that the model's factual commitments are more reliable. This demonstrates an optimized trade-off between truthfulness and informativeness, effectively mitigating the risk of misinformation.

**Resolution of Fixed-Reward Brittleness.** A comparative analysis highlights a fundamental failure mode in the GRPO-UC baselines: extreme sensitivity to the fixed reward $r_u$. While specific static rewards may yield balanced results for isolated datasets, they fail to generalize across varying difficulty gradients and task types. For instance, on Llama-3.1-8B-Instruct, a high $r_u \geq 0.5$ in Math & Text Reasoning tasks inevitably triggers reward hacking, where the model bypasses the cognitive cost of reasoning in favor of guaranteed uncertainty rewards, leading to a catastrophic collapse in F1 scores (Notably, while uncertainty may converge to 100% during training, the test set ratio might not reach the same level due to the lack of perfect identical

*Table 3.* Ablation study results on the Llama-3.1-8B-Instruct model. We evaluate the contribution of Ternary Advantage Decoupling (TAD), Dynamic Uncertainty Reward Adjustment (DURA), Non-Ternary Filtering (NTF), and Low-Resource Extensions (LRE).

| TAD | DURA | NTF | LRE | Math & Text Reasoning | | | General Tasks | | |
|---|---|---|---|---|---|---|---|---|---|
| | | | | Uncertainty | PAQ | F1 | Uncertainty | PAQ | F1 |
| × | ✓ | ✓ | ✓ | 50.33 | 22.56 | 16.21 | 12.40 | 51.17 | 48.63 |
| ✓ | × | ✓ | ✓ | 79.91 | 35.22 | 13.16 | 80.31 | 58.96 | 23.41 |
| ✓ | ✓ | × | ✓ | 37.96 | 28.51 | 22.93 | 43.38 | 55.68 | 43.05 |
| ✓ | ✓ | ✓ | × | 43.19 | 27.83 | 21.12 | 52.95 | 57.38 | 39.99 |
| ✓ | ✓ | ✓ | ✓ | 39.09 | 28.45 | 22.65 | 47.33 | 58.58 | 43.10 |

*Table 4.* Parameter sensitivity for Llama-3.1-8B-Instruct on General Tasks. For the $w$ study, $G = 8$; for the $G$ study, $w = 1$.

| Setting | Value | GPQA-Diamond | | MMLU-Redux2 | | Average | |
|---|---|---|---|---|---|---|---|
| | | PAQ | F1 | PAQ | F1 | PAQ | F1 |
| | 0.5 | 36.54 | 10.89 | 80.72 | 64.93 | 58.63 | 37.91 |
| $w$ | 1.0 | 36.02 | 17.18 | 81.13 | 69.03 | 58.58 | 43.10 |
| | 2.0 | 31.09 | 19.28 | 77.96 | 70.82 | 54.53 | 45.05 |
| | 4 | 33.46 | 20.54 | 76.69 | 65.20 | 55.08 | 42.87 |
| $G$ | 8 | 36.02 | 17.18 | 81.13 | 69.03 | 58.58 | 43.10 |
| | 16 | 34.75 | 25.95 | 80.69 | 67.83 | 57.72 | 46.89 |

distribution, though it remains significantly high despite the distribution shift.). Conversely, in General Tasks, a lower $r_u = 0.2$ results in the suppression of uncertainty learning; the weak incentive fails to encourage the model to acknowledge its knowledge boundaries, leaving it prone to random guessing and severe overconfident errors. UCPO's dynamic mechanism addresses these issues by preventing avoidance degeneracy in complex reasoning while eliciting honest expressions in general knowledge tasks, thereby maintaining robust problem-solving utility.

**Metric Behavior Analysis.** Although UCPO achieves strong aggregate performance, improvements in Accuracy or F1 need not be uniform across all Math & Text Reasoning datasets. One reason is that Qwen3-8B has already been strongly trained on mathematical reasoning, leaving limited room for additional RL alignment to further improve deterministic correctness. In such settings, standard GRPO may show little improvement over Prompt-UC, or even slight decreases in some cases; similar limited-gain behavior after RL alignment has also been observed in prior RL-based optimization studies (Liu et al., 2026). Thus, UCPO's gains should be read mainly as improved uncertainty calibration and more reliable answer commitments, reflected by PAQ, rather than stronger reasoning ability. In General Tasks, lower F1 can occur because multiple-choice formats allow answers made under uncertainty to be scored as correct by chance. UCPO tends to convert these unreliable answers into explicit uncertainty, which may reduce coverage but improves the reliability of the remaining answered responses. Consistently, UCPO substantially im-

proves PAQ. The uncertain-subset analysis in Appendix D.1 further supports this interpretation: when evaluated on cases that UCPO marks as uncertain, GRPO remains prone to overconfident erroneous answers, indicating that UCPO's uncertainty outputs reflect genuine knowledge boundaries rather than arbitrary abstention.

## 4.4. Ablation Study

To quantify each component's contribution in UCPO, we conduct ablation studies on Llama-3.1-8B-Instruct (Table 3). Non-Ternary Filtering (NTF) refers to filtering out samples missing either correct or incorrect rollouts, as defined in the main text, while Low-Resource Extensions (LRE) are detailed in the supplementary material.

Removing Ternary Advantage Decoupling (TAD) significantly degrades PAQ; specifically in simpler tasks, the absence of decoupling allows deterministic gradients to overshadow subtle calibration signals, hindering the acquisition of effective uncertainty representations. Regarding the reward mechanism, omitting DURA leads to a performance collapse across both domains as the model over-optimizes for uncertainty rewards (a reward-hacking surge to ∼80% uncertainty). Moreover, the exclusion of Non-Ternary Filtering (NTF) induces training fluctuations and suboptimal convergence, whereas its inclusion, alongside Low-Resource Extensions (LRE), consistently yields higher F1 and PAQ scores by providing the necessary robustness for calibration under data constraints. These findings are further corroborated by the training trajectories in Figure 9 (Appendix), highlighting the synergistic effect of these modules in balancing accuracy and metacognitive calibration.

## 4.5. Parameter Sensitivity

DURA uses $w$ to control the range and balance of $\gamma(q)$. With sufficient rollouts, $\gamma(q) \in (-w, 1)$; when $P_r = P_w = P_u = 1/3$, the default $w = 1$ yields $\gamma(q) = 0$, giving a neutral equilibrium among correct, incorrect, and uncertain responses. Table 4 shows that $w$ mainly controls the trade-off between PAQ and F1: smaller values preserve higher reliability, whereas larger values improve F1 at the cost of PAQ. For the rollout group size, $G = 8$ achieves the best

average PAQ, while $G = 16$ further improves average F1, indicating that larger groups provide more stable advantage estimates than $G = 4$.

## 5. Related Work

### 5.1. Reinforcement Learning

Reinforcement Learning (RL) has emerged as the cornerstone paradigm for aligning Large Language Models and enhancing their multi-step reasoning capabilities (Li et al., 2025; Zhang et al., 2025). Within this landscape, GRPO represents a significant milestone; by introducing a critic-free architecture and group-relative normalization, it substantially reduces training variance and accelerates convergence (Shao et al., 2024; Guo et al., 2025). Building upon this foundation, subsequent studies such as Dr.GRPO (Liu et al., 2025a), DAPO (Yu et al., 2025), and LitePPO (Liu et al., 2025b) have further refined training stability through sophisticated sampling strategies, clipping mechanisms, and loss function optimizations. Meanwhile, PSR-NSR (Zhu et al., 2025) and NGRPO (Nan et al., 2025) explore the exploitation of negative samples to extract meaningful learning signals from erroneous responses. Diverging from these approaches, our proposed UCPO introduces uncertainty modeling to prevent the model from compulsive fitting of incomprehensible error signals, thereby reducing overconfident erroneous assertions during the training process.

### 5.2. Uncertainty-aware Learning in LLMs

Current LLMs frequently exhibit overconfident factual errors, largely due to underdeveloped refusal mechanisms (Wen et al., 2025; Madhusudhan et al., 2025; Kirichenko et al., 2025). To empower models to articulate uncertainty, early research primarily utilized uncertainty-guided prompts or strategies to induce verbalized confidence for calibration (Slobodkin et al., 2023; Tian et al., 2023). Moving further, the Supervised Fine-Tuning paradigm introduced specialized datasets with uncertainty annotations to map knowledge boundaries (Amayuelas et al., 2024; Kapoor et al., 2024; Cheng et al., 2024). While fine-tuning helps identify unanswerable queries, it is constrained by high labeling costs and an inability to capture a model's internal limits. Recent RL-based reasoning advances have reduced supervision needs, but binary rewards often encourage overconfident guessing over honest uncertainty (Kalai et al., 2025; Yao et al., 2025). To mitigate this, TruthRL (Wei et al., 2025) and KnowRL (Ren et al., 2025) treat uncertain states as fixed intermediate reward values to guide the model toward exploring its knowledge limits. Despite their success in specific tasks, their hyperparameter sensitivity often triggers an imbalance between over-refusal and erroneous assertion. Unlike fixed-reward methods, UCPO dynamically adjusts uncertainty rewards to ensure robust learning of uncertainty expressions across diverse tasks.

Relatedly, uncertainty quantification has been studied through inference-time confidence or semantic uncertainty estimation (Malinin & Gales, 2020; Kuhn et al., 2023; Ielanskyi et al., 2025) and RL-based calibration that trains models to report numerical confidence after reasoning (Damani et al., 2025). UCPO differs by optimizing uncertainty expression as a discrete abstention policy rather than a calibrated confidence score, prioritizing alignment with the general reasoning paradigm where the model autonomously decides whether to express uncertainty during end-to-end reasoning instead of relying on external confidence thresholds.

## 6. Conclusion

This paper presents UCPO, a reinforcement learning framework designed to empower LLMs to express uncertainty when encountering unknown queries. By implementing Ternary Advantage Decoupling and Dynamic Uncertainty Reward Adjustment, UCPO resolves the gradient interference and reward hacking inherent in naive methods, achieving stable and effective uncertainty learning. A remaining limitation is that the distribution ratios of different rollout types potentially influence uncertainty learning, a phenomenon observed in our experiments but not fully explored. Future work will investigate the specific impact of ternary signal distributions on training dynamics and explore more effective balancing strategies.

## Impact Statement

This work aims to improve LLM reliability by training models to express uncertainty instead of making overconfident erroneous assertions. It may reduce misleading outputs in high-stakes settings, but UCPO does not guarantee detection of all unknown queries and should not replace human or expert review. Practical deployment should pair UCPO with task-specific risk assessment, oversight, and external verification to manage both over-abstention and residual errors.

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

## A. Formal Analysis of Ternary Imbalance

**Setup.** Let a rollout group for prompt $q$ contain proportions $P_r$, $P_w$, and $P_u$ of right, wrong, and uncertain outputs, with

$$P_r + P_w + P_u = 1, \qquad P_r, P_w, P_u \geq 0. \tag{6}$$

Fixed-reward GRPO-UC assigns rewards

$$r_{\text{right}} = 1, \qquad r_{\text{wrong}} = 0,$$
$$r_{\text{u}} = r_u, \qquad r_u \in (0, 1). \tag{7}$$

The group reward mean and variance are

$$\mu = P_r + r_u P_u, \qquad \sigma^2 = P_r + r_u^2 P_u - (P_r + r_u P_u)^2. \tag{8}$$

We assume $\sigma > 0$ in the following derivation.

**High-performance uncertainty suppression.** The normalized advantage of an uncertain rollout is

$$A_u^{\text{GRPO-UC}} = \frac{r_u - \mu}{\sigma} = \frac{r_u(1 - P_u) - P_r}{\sigma}. \tag{9}$$

Since $\sigma > 0$ and $1 - P_u = P_r + P_w$, the uncertain advantage becomes negative exactly when

$$r_u(1 - P_u) - P_r < 0 \iff r_u(P_r + P_w) < P_r$$
$$\iff P_r > \frac{r_u}{1 - r_u} P_w. \tag{10}$$

Thus, as the model improves and $P_w$ decreases, the threshold in Eq. (10) approaches zero. In the high-performance region, even locally rational uncertainty can receive negative advantage simply because the global group mean is high.

**Low-performance reward-hacking dominance.** Let the aggregate right and uncertain advantage contributions be

$$C_r = GP_r \frac{1 - \mu}{\sigma}, \qquad C_u = GP_u \frac{r_u - \mu}{\sigma}, \tag{11}$$

and define the net right advantage as $\text{NRA} = C_r - C_u$. Then $\text{NRA} < 0$, meaning uncertain gradients dominate right gradients, if and only if

$$P_r \big[ P_w + P_u(2 - r_u) \big] < P_u r_u(1 - P_u). \tag{12}$$

To see this, $\text{NRA} < 0$ is equivalent to

$$P_r(1 - \mu) < P_u(r_u - \mu). \tag{13}$$

Substituting $\mu = P_r + r_u P_u$ gives

$$P_r \big[ P_w + P_u(1 - r_u) \big] < P_u \big[ r_u(1 - P_u) - P_r \big], \tag{14}$$

which rearranges to Eq. (12).

The condition in Eq. (12) also exposes how reward hacking is triggered in low-performance regimes. Using $P_w = 1 - P_u - P_r$, it can be written as

$$P_r \big[ 1 + P_u(1 - r_u) - P_r \big] < r_u P_u(1 - P_u). \tag{15}$$

For any fixed $P_u \in (0, 1)$ and $r_u > 0$, the right side of Eq. (15) is positive. Hence, for sufficiently small but nonzero $P_r$, fixed-reward GRPO-UC enters an uncertainty-dominant regime: the aggregate gradient contribution from uncertain rollouts exceeds that from right rollouts.

This regime is not a static imbalance but a self-reinforcing cascade. Without setting $P_r = 0$, the uncertain advantage is

$$A_u^{\text{GRPO-UC}} = \frac{r_u(1 - P_u) - P_r}{\sigma}, \tag{16}$$

which remains positive whenever

$$P_r < r_u(1 - P_u). \tag{17}$$

In the same regime, wrong rollouts have advantage $A_w = -\mu/\sigma < 0$, so the policy is pushed away from wrong answers and toward the positively rewarded uncertainty path, while the right-answer signal is weak because its mass $P_r$ is small. Equations (15) and (17) also clarify the role of $r_u$: a larger fixed uncertainty reward enlarges both the NRA trigger region and the region where uncertain rollouts keep positive advantage. Reducing $r_u$ can delay the cascade, but it does not introduce a negative-feedback mechanism.

To characterize the early cascade, expand the variance for $P_r \ll 1$:

$$\sigma^2 = r_u^2 P_u(1 - P_u) + P_r(1 - 2r_u P_u) - P_r^2. \tag{18}$$

Thus, on interior intervals of $P_u \in (0, 1)$,

$$A_u^{\text{GRPO-UC}} = \sqrt{\frac{1 - P_u}{P_u}} + O(P_r),$$
$$H_{\text{GRPO-UC}}(P_u; P_r) \triangleq P_u A_u^{\text{GRPO-UC}} \tag{19}$$
$$= \sqrt{P_u(1 - P_u)} + O(P_r).$$

Here $H_{\text{GRPO-UC}}$ denotes the aggregate uncertainty pressure per rollout, i.e., the uncertain-rollout advantage weighted by the proportion of uncertain outputs; equivalently, $H_{\text{GRPO-UC}} = C_u/G$. The leading-order pressure $H_0(P_u) = \sqrt{P_u(1 - P_u)}$ satisfies

$$\frac{dH_0}{dP_u} = \frac{1 - 2P_u}{2\sqrt{P_u(1 - P_u)}}. \tag{20}$$

Therefore, for $P_u < 1/2$ and sufficiently small $P_r$, an initial increase in $P_u$ increases the aggregate uncertainty pressure, which further strengthens the gradient toward uncertain responses. This gives the acceleration stage of reward hacking:

$$P_u \uparrow \implies H_{\text{GRPO-UC}} \uparrow \implies P_u \uparrow. \tag{21}$$

After $P_u$ passes $1/2$, the leading-order aggregate pressure decreases, but the individual uncertain advantage remains

positive as long as Eq. (17) holds. Since there is no explicit anti-hacking term that turns the uncertainty channel negative at an interior equilibrium, the policy can remain locked onto the uncertainty shortcut. As training suppresses non-uncertain paths and $P_r$ stays small or decreases, the positivity condition in Eq. (17) moves toward the boundary, which can drive a cascade toward the all-uncertainty collapse $P_u \to 1$. The rapid rise of GRPO-UC uncertainty ratios in Figure 4 is the empirical counterpart of this mechanism.

**UCPO feedback equilibrium.** UCPO removes the fixed uncertainty reward from the uncertain advantage. For NTF-valid groups with $P_r, P_w > 0$, define the deterministic conditional right ratio

$$\tilde{P}_r = \frac{P_r}{P_r + P_w} \in (0, 1). \qquad (22)$$

For clarity, the sign analysis below omits the numerical stabilizer in the deterministic normalization; adding it only rescales the positive anchor. The deterministic right-rollout anchor is

$$\hat{A}_{right} = \frac{1 - \tilde{P}_r}{\sqrt{\tilde{P}_r(1 - \tilde{P}_r)}} = \sqrt{\frac{1 - \tilde{P}_r}{\tilde{P}_r}} > 0. \qquad (23)$$

Consistent with the uncertainty channel in the main text, UCPO defines the uncertain rollout advantage as

$$\hat{A}_{i,t}^{unc} = \gamma(q)\hat{A}_{right}, \qquad (24)$$

where the implementation uses the stabilized DURA gain

$$\gamma_\epsilon(q) = \frac{P_w}{P_u + P_w + \epsilon}(1 - P_u) - w\frac{P_r}{P_r + P_w + \epsilon}P_u. \qquad (25)$$

For the analytic sign argument below, we study the idealized population-level form obtained by taking $\epsilon \to 0$ on NTF-valid groups, where the denominators are positive:

$$\gamma(q) = \frac{P_w}{P_u + P_w}(1 - P_u) - w\frac{P_r}{P_r + P_w}P_u, \qquad w > 0. \qquad (26)$$

To analyze the anti-hacking equilibrium without leaving the probability simplex, fix the deterministic correctness ratio

$$\kappa = \frac{P_r}{P_r + P_w} \in (0, 1) \qquad (27)$$

and vary the uncertainty mass $s = P_u$ along

$$\begin{aligned} P_u &= s, \qquad s \in [0, 1), \\ P_r &= \kappa(1 - s), \qquad (28) \\ P_w &= (1 - \kappa)(1 - s). \end{aligned}$$

Along this path, $\tilde{P}_r = \kappa$ and $\hat{A}_{right} = \sqrt{(1 - \kappa)/\kappa} > 0$. Substituting Eq. (28) into Eq. (26) yields, for $s < 1$,

$$\gamma(s; \kappa) = \frac{(1 - \kappa)(1 - s)^2}{1 - \kappa + \kappa s} - w\kappa s. \qquad (29)$$

The boundary values are

$$\gamma(0; \kappa) = 1, \qquad \lim_{s \to 1^-} \gamma(s; \kappa) = -w\kappa < 0. \qquad (30)$$

Moreover,

$$\begin{aligned} \frac{\partial\gamma(s; \kappa)}{\partial s} = &-\frac{(1 - \kappa)(1 - s)}{(1 - \kappa + \kappa s)^2} \\ &\times \left[2(1 - \kappa + \kappa s) + \kappa(1 - s)\right] - w\kappa < 0. \end{aligned} \qquad (31)$$

Therefore, by continuity on $[0, 1)$, the negative left-boundary limit, and strict monotonicity, there exists a unique $s^\star \in (0, 1)$ such that $\gamma(s^\star; \kappa) = 0$.

Because $\hat{A}_{right} > 0$, Eq. (24) shows that the sign of $\hat{A}_{i,t}^{unc}$ is the sign of $\gamma$. Consequently,

$$\lim_{s \to 0} \hat{A}_{i,t}^{unc}(s) = \hat{A}_{right} > 0, \qquad (32)$$

which gives anti-suppression: uncertainty remains learnable when the model rarely expresses it. This explicitly mitigates high-performance uncertainty suppression: even when the deterministic correctness ratio $\kappa$ is high, the low-uncertainty limit keeps $\hat{A}_{i,t}^{unc}$ positive instead of letting a high global reward mean turn uncertainty into a negative-advantage action; its magnitude naturally decreases as deterministic correctness approaches certainty. For $s > s^\star$,

$$s\hat{A}_{i,t}^{unc}(s) = s\gamma(s; \kappa)\hat{A}_{right} < 0, \qquad (33)$$

which gives anti-hacking feedback: once uncertainty saturates beyond the equilibrium, further uncertain outputs receive negative aggregate pressure. These two properties formalize the balanced ternary regions illustrated in Figure 2(c,d).

## B. UCPO Extensions for Low-Resource Scenarios

In low-resource settings with restricted rollout numbers (e.g., $G \in [4, 8]$), the sparse rollout space introduces significant statistical bias and high variance. The bias arises from the coarse granularity of the probability estimates; with so few rollouts, the discrete nature of the ratios prevents $\gamma(q)$ from spanning its full theoretical range.

As illustrated in Figure 6, the effective uncertainty-channel gain values of $\gamma(q)$ for $G = 8$ are constrained to $[-0.354, 0.732]$, a range significantly narrower than the theoretical extrema of $\pm 1$ under the implemented setting

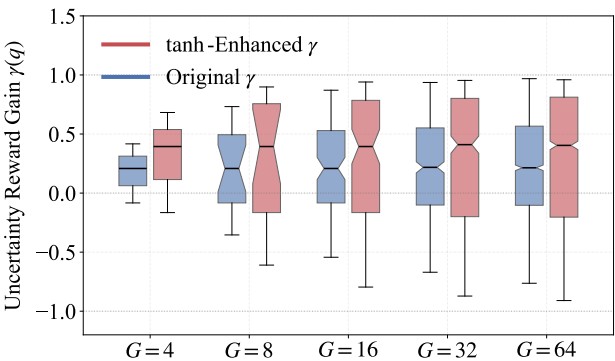

*Figure 6.* Impact of rollout group size $G$ on the statistical distribution of gain $\gamma(q)$.

$w = 1$. Here, the effective range is computed over NTF-valid groups that contain at least one uncertain rollout, since $\gamma(q)$ affects the loss only through $\hat{A}_{i,t}^{unc}$ and has no training effect when $P_u = 0$. Note that while broader values are mathematically possible, this empirical range results from applying Non-Ternary Filtering (NTF), which excludes cases where $P_r$ or $P_w$ equals zero, together with the requirement that the uncertainty channel is active. Such compression diminishes the reward's discriminative power and weakens the model's capacity to correct persistent overconfident errors. Simultaneously, the limited rollout size amplifies sensitivity to individual outcomes, leading to high variance that destabilizes advantage estimation and often masks the underlying gradient. To restore the incentive's dynamic range and ensure training stability, we extend the framework with the following strategies:

**Batch-Level Smoothing Fusion** To mitigate high variance caused by minimal group sizes, we introduce a weighted fusion mechanism that incorporates batch-level priors:

$$\gamma_{\text{fused}}(q) = \lambda \cdot \gamma_{\text{sample}}(q) + (1 - \lambda) \cdot \bar{\gamma}_{\text{batch}} \quad (34)$$

where $\gamma_{\text{sample}}(q)$ represents the instance-specific gain and $\bar{\gamma}_{\text{batch}}$ denotes the mean gain across the current training batch. The hyperparameter $\lambda \in [0, 1]$ controls the trade-off between individualized response and global stability; in our implementation, we set $\lambda = 0.5$ to achieve a balanced smoothing effect. This design effectively suppresses stochastic fluctuations while preserving the necessary sample-specific nuances required for fine-grained policy optimization.

**Non-linear Mapping Enhancement** To overcome the incentive bottleneck, we apply a hyperbolic tangent transformation to stretch the dynamic range of the gain:

$$\gamma_{\text{final}}(q) = \tanh\left(\alpha \cdot \gamma_{\text{fused}}(q)\right) \quad (35)$$

By pushing moderate gains toward the $\pm 1$ boundaries, this mapping restores the reward signal's ability to distinguish

between varying rollout qualities, thereby overcoming a limitation frequently encountered in small-rollout settings. In our implementation with $G = 8$, we set the scaling factor $\alpha = 2$ to effectively recalibrate the incentive range and maintain sufficient optimization pressure.

These extensions collectively allow the framework to maintain high feedback intensity and decision robustness even under severe resource constraints.

## C. Prompting Details

> {question}
> Please reason step by step. If confident based on reliable knowledge, provide a clear answer and box it with \\boxed{}.
> If the question lacks clarity, exceeds your knowledge, involves speculation, prediction, opinion, or any uncertainty, do not guess. State your limitation and output \\boxed{uncertainty}.

*Figure 7.* Prompt template used in Math & Text Reasoning tasks for open-ended question answering.

> {question}
> Please reason step by step. If you are confident based on reliable knowledge, only output the choice letter in the answer field, e.g., answer: C.
> If the question lacks clarity, exceeds your knowledge, involves speculation, prediction, opinion, or any uncertainty, do not guess. Instead, state your limitation and output uncertainty, e.g., answer: uncertainty.

*Figure 8.* Prompt template designed for General Tasks involving multiple-choice questions.

Figure 7 shows the prompt template used for Math & Text Reasoning tasks, where the model is guided to generate step-by-step reasoning for open-ended questions. Figure 8 illustrates the prompt design for General Tasks, specifically multiple-choice questions, where the model selects the correct answer from given options.

**Uncertainty identification.** All experiments use fixed output formats and classify responses through rule-based or string-matching checks. If a response contains vague or partial uncertainty but does not follow the required uncertainty format, we treat it as a definitive answer and judge it as correct or incorrect. We do not use a reward model

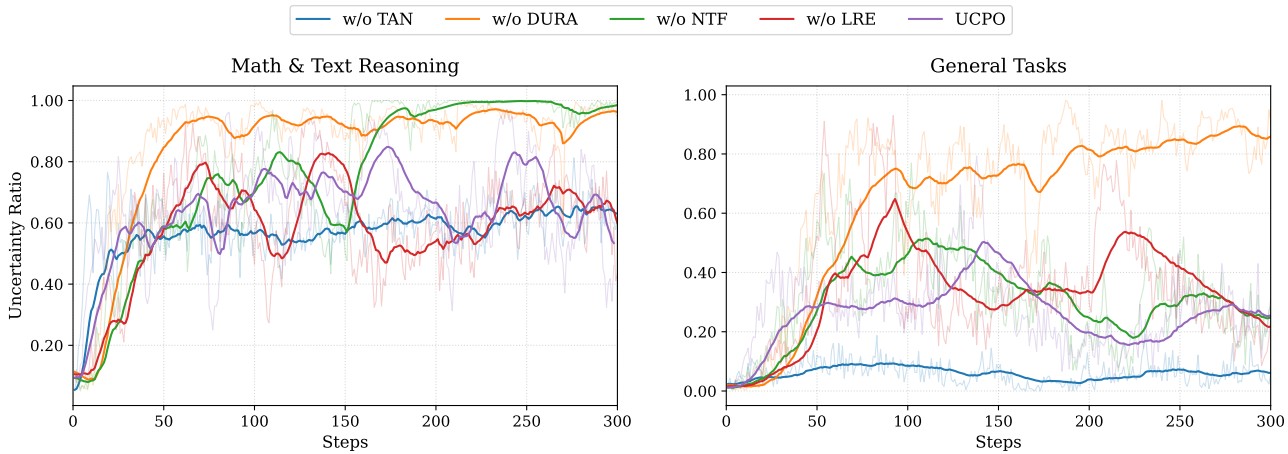

*Figure 9.* Training dynamics of UCPO ablation variants on Llama-3.1-8B-Instruct.

in the current experiments; UCPO is decoupled from the classifier, so open-ended settings can replace these rules with a learned reward model.

## D. Additional Experimental Results

We organize the supplementary experiments around three questions: whether UCPO's uncertainty predictions reflect meaningful knowledge boundaries, whether UCPO training remains stable, and whether the framework extends beyond the main experimental setting.

### D.1. Validity of Uncertainty Predictions

To assess whether UCPO's uncertainty predictions correspond to genuine knowledge boundaries rather than excessive refusal, we evaluate a model without uncertainty training on the subset of cases marked uncertain by UCPO. If these predictions are meaningful, this subset should be substantially harder than the full evaluation set even for GRPO. Using Qwen3-8B, we therefore measure GRPO accuracy on UCPO-uncertain cases. As shown in Table 5, GRPO accuracy on this subset is much lower than its full-set accuracy, indicating that UCPO identifies low-confidence cases rather than arbitrary abstention.

*Table 5.* GRPO accuracy on the full evaluation set and the UCPO-uncertain subset for Qwen3-8B on General Tasks.

| Dataset | Full Acc. | UCPO-Uncertain Acc. | Drop |
|---|---|---|---|
| GPQA-Diamond | 59.35 | 36.96 | -22.39 |
| MMLU-Redux2 | 87.62 | 55.08 | -32.54 |
| Average | 73.48 | 46.02 | -27.46 |

The 46.02% average accuracy on UCPO-uncertain cases, compared with 73.48% on the full set, supports that these samples are genuinely uncertain for the model. For multiple-

choice questions, random guessing among four options yields 25%, while guessing after eliminating two options yields 50%. The observed 46.02% accuracy is consistent with low-confidence cases near the model's knowledge boundary. These results suggest that at least part of the observed F1 trade-off may come from converting low-confidence or incidental correct answers into uncertainty outputs, which aligns with UCPO's design goal of favoring more reliable factual commitments over chance correctness.

### D.2. Comparative Analysis of Ablation Study Training Processes

The training trajectories in Figure 9 provide a supplementary dynamic analysis to the ablation results presented in Section 4.4, revealing the mechanistic synergy required for stable optimization. A critical failure mode occurs upon removing DURA, where the model suffers from severe reward-hacking. In Math and Text Reasoning tasks, the model rapidly converges toward a degenerate state of near 100% uncertainty prediction to maximize reward signals. Conversely, the absence of TAD leads to erratic convergence in General Tasks because deterministic gradients overshadow the subtle signals required for calibration, causing the model to neglect uncertainty estimation entirely.

The inclusion of NTF serves as a vital regularizer for the loss landscape. While variants lacking NTF may initially converge in General Tasks, they exhibit catastrophic collapse during the later stages of Math and Text Reasoning training, highlighting the instability introduced by noisy, non-ternary samples. Regarding LRE, its contribution to long-term stability appears less definitive compared to other modules, which can be attributed to inherent variances in sample distribution and fluctuating task difficulty during the extension process. It should be noted that the scaling factors within LRE facilitate a perceptibly faster convergence in un-

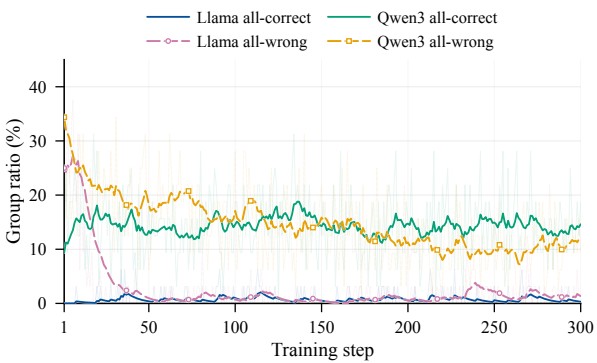

*Figure 10.* NTF-related homogeneous response group ratios over the first 300 training steps. All-correct/all-wrong denote groups whose rollouts are uniformly correct/incorrect. Translucent traces show raw batch-level ratios; darker traces show EMA-smoothed trends.

certainty learning by amplifying relevant advantage signals. The complex interplay between these scaling factors and data diversity warrants further exploration in future work to optimize the efficiency of metacognitive alignment.

### D.3. NTF Filtering Ratio and Sample Efficiency

To quantify the scope of NTF, we track the ratios of homogeneous response groups during training. An all-correct group contains only correct rollouts, and an all-wrong group contains only incorrect rollouts. Such groups provide little ternary contrast for learning when to express uncertainty, similar to the zero-advantage cases in standard GRPO. The statistics in this section are computed on DAPO-Math-17k training batches and therefore describe training-time filtering behavior rather than evaluation-set composition.

Figure 10 shows that the filtered portion remains limited and becomes cleaner as training proceeds. For Llama-3.1-8B-Instruct, all-wrong groups are common at the beginning but rapidly vanish, while all-correct groups remain rare, fluctuating around roughly 1% during training. For Qwen3-8B, the all-wrong ratio also declines, whereas the all-correct ratio stays in a moderate band. Since Qwen3-8B has already undergone RL on math problems, here we apply an additional preprocessing step that filters simple all-correct samples before rollout generation, reducing ineffective reasoning on examples that are already solved reliably. Thus, the remaining all-correct groups are not merely trivial instances.

Together with the ablation trajectories in Figure 9, these results indicate that NTF does not discard the dominant source of optimization signal. Instead, it removes a bounded subset of low-signal groups while preserving mixed-response groups that still carry informative ternary contrast, which explains why NTF improves training stability without sacri-

ficing sample efficiency.

### D.4. Multi-seed Training Stability

To assess robustness to stochastic training variation, rather than variability from multiple responses sampled for the same instance, we repeat UCPO training for Llama-3.1-8B-Instruct in the General Tasks setting under three independent random seeds. Table 6 shows low variance in PAQ and F1, indicating stable convergence across random-seed runs.

*Table 6.* UCPO performance across three independent random-seed runs for Llama-3.1-8B-Instruct in the General Tasks setting.

| Metric | Seed 1 | Seed 2 | Seed 3 | Mean ± Std |
|--------|--------|--------|--------|-------------|
| PAQ | 58.58 | 59.00 | 57.61 | 58.40 ± 0.72 |
| F1 | 43.10 | 45.83 | 43.94 | 44.29 ± 1.40 |

### D.5. Real Unanswerable Queries

We evaluate transfer to explicitly unanswerable queries using AbstentionBench (Kirichenko et al., 2025). To adapt the evaluation to the constrained-choice format used in General Tasks training, we select the multiple-choice subsets from AbstentionBench: GPQA-Diamond-Abstain, derived from GPQA-Diamond (Rein et al., 2024); Big-Bench-Disambiguate, derived from BIG-Bench (Srivastava et al., 2023); MMLU-Math, derived from the mathematics subsets of MMLU (Hendrycks et al., 2020); MediQ (Li et al., 2024); and MoralChoice (Scherrer et al., 2023). These subsets preserve answerable examples while including should-abstain samples introduced or curated by AbstentionBench. In particular, the GPQA-Diamond and MMLU-Math variants remove necessary context from originally answerable questions, MediQ contains clinical cases with insufficient patient information, and the Disambiguate and MoralChoice subsets contain prompts where the intended referent or morally preferred choice is not uniquely determined.

Table 7 shows that GRPO has zero abstain recall across all selected datasets, indicating that standard RL strongly favors definitive answers even when the prompt lacks sufficient evidence. GRPO-UC improves abstention recall but is less stable across domains, particularly on Big-Bench-Disambiguate and MediQ. UCPO achieves the strongest macro-average total accuracy, abstain recall, and abstain precision, with especially large gains on GPQA-Diamond-Abstain and MMLU-Math. These results suggest that UCPO's learned uncertainty behavior transfers beyond synthetic training labels to real should-abstain cases.

### D.6. Larger Model Extension

Table 8 examines whether UCPO remains effective across different model sizes by comparing Qwen3-8B and Qwen3-

*Table 7.* Performance on multiple-choice real unanswerable queries from AbstentionBench.

| Dataset | Methods | Total Acc. | Abstain Recall | Abstain Prec. |
|---|---|---|---|---|
| GPQA-Diamond-Abstain | Prompt-UC | 21.67 | 15.00 | 60.00 |
| | GRPO | 22.50 | 0.00 | 0.00 |
| | GRPO-UC$^{\dagger 0.5}$ | 43.75 | 70.00 | 60.87 |
| | UCPO | 47.50 | 82.50 | 58.93 |
| Big-Bench-Disambiguate | Prompt-UC | 45.60 | 12.00 | 39.13 |
| | GRPO | 58.40 | 0.00 | 0.00 |
| | GRPO-UC$^{\dagger 0.5}$ | 49.60 | 22.67 | 31.48 |
| | UCPO | 50.00 | 57.33 | 33.59 |
| MMLU-Math | Prompt-UC | 48.12 | 36.09 | 96.00 |
| | GRPO | 35.34 | 0.00 | 0.00 |
| | GRPO-UC$^{\dagger 0.5}$ | 65.41 | 83.46 | 73.03 |
| | UCPO | 75.94 | 85.71 | 83.21 |
| MediQ | Prompt-UC | 36.23 | 2.56 | 75.00 |
| | GRPO | 36.66 | 0.00 | 0.00 |
| | GRPO-UC$^{\dagger 0.5}$ | 44.85 | 19.30 | 67.33 |
| | UCPO | 53.66 | 44.14 | 64.86 |
| MoralChoice | Prompt-UC | 49.01 | 0.00 | 0.00 |
| | GRPO | 49.74 | 0.00 | 0.00 |
| | GRPO-UC$^{\dagger 0.5}$ | 51.43 | 3.82 | 100.00 |
| | UCPO | 65.91 | 32.94 | 96.55 |
| Macro Avg. | Prompt-UC | 40.13 | 13.13 | 54.03 |
| | GRPO | 40.53 | 0.00 | 0.00 |
| | GRPO-UC$^{\dagger 0.5}$ | 51.01 | 39.85 | 66.54 |
| | UCPO | 58.60 | 60.52 | 67.43 |

32B in the General Tasks setting. UCPO improves average PAQ at both parameter scales, from 71.96 to 79.68 on Qwen3-8B and from 76.88 to 87.07 on Qwen3-32B, while maintaining comparable average F1. These results indicate that the calibration benefit of UCPO is not limited to a single model size, but extends consistently across 8B and 32B models.

*Table 8.* UCPO effectiveness across Qwen3 model sizes in the General Tasks setting.

| Methods | GPQA-Diamond | | MMLU-Redux2 | | Average | |
|---|---|---|---|---|---|---|
| | PAQ | F1 | PAQ | F1 | PAQ | F1 |
| *Qwen3-8B* | | | | | | |
| Prompt-UC | 56.84 | 55.67 | 87.08 | 86.33 | 71.96 | 71.00 |
| UCPO | 67.70 | 56.16 | 91.67 | 85.42 | 79.68 | 70.79 |
| *Qwen3-32B* | | | | | | |
| Prompt-UC | 63.95 | 63.62 | 89.81 | 89.27 | 76.88 | 76.44 |
| UCPO | 79.80 | 64.80 | 94.33 | 87.14 | 87.07 | 75.97 |

### D.7. Inference-time Scaling of Uncertainty Expression

Table 9 explores whether UCPO's uncertainty expressions can also serve as useful signals at inference time. For each test instance, vote@$n$ samples $n$ responses and aggregates the final prediction by majority voting. UCPO vote@$n$ applies the same aggregation rule to UCPO responses, while UCPO any-uncertain@$n$ uses a more conservative rule: if any sampled response expresses uncertainty, the final output is treated as uncertain; otherwise, majority voting is applied to the remaining definitive answers. We report PAQ and F1 because they capture complementary aspects of inference-time behavior. PAQ measures the reliability of answered responses, whereas F1 reflects the balance between providing useful answers and avoiding overconfident errors.

The results show that UCPO already improves answer reliability at $n = 1$, increasing PAQ from 74.27 under GRPO to 81.12. With standard voting, increasing the number of samples further raises UCPO PAQ to 84.05 at $n = 16$, while F1 remains broadly comparable. The conservative any-uncertain@$n$ rule yields the strongest reliability gain, reaching 90.86 PAQ at $n = 16$, but its F1 decreases because a single uncertain sample can convert an otherwise answerable instance into abstention. These trends indicate that uncertainty expressions provide an effective inference-time risk signal: they can be exploited to improve precision when reliability is prioritized, with the expected trade-off of reduced answer coverage.

*Table 9.* Inference-time scaling of uncertainty-aware aggregation for Qwen3-8B in the General Tasks setting.

| Method | $n = 1$ | | $n = 4$ | | $n = 8$ | | $n = 16$ | |
|---|---|---|---|---|---|---|---|---|
| | PAQ | F1 | PAQ | F1 | PAQ | F1 | PAQ | F1 |
| GRPO (vote@$n$) | 74.27 | 73.91 | 77.70 | 75.43 | 76.83 | 75.80 | 76.32 | 75.84 |
| UCPO (vote@$n$) | 81.12 | 71.78 | 82.45 | 70.23 | 83.25 | 72.23 | 84.05 | 73.02 |
| UCPO (any-uncertain@$n$) | 81.12 | 71.78 | 82.45 | 70.23 | 87.76 | 70.78 | 90.86 | 68.31 |

### D.8. Compatibility with Diverse RL Methods

UCPO is a framework-agnostic approach that can be seamlessly integrated with advanced RL methods. To demonstrate this, we extend UCPO to DAPO, which utilizes decoupled clipping and dynamic sampling to mitigate entropy collapse and enhance exploration (Yu et al., 2025). As illustrated in Table 10, we evaluate the compatibility across three comparative groups on Llama-3.1-8B-Instruct.

*Table 10.* Performance on Llama-3.1-8B-Instruct across general tasks. Here, DAPO-UC denotes the direct addition of uncertainty as a reward term to the DAPO baseline, while DAPO-UCPO represents the full integration of our proposed UCPO framework with DAPO.

| Methods | GPQA-Diamond | | MMLU-Redux2 | | Average | |
|---|---|---|---|---|---|---|
| | PAQ | F1 | PAQ | F1 | PAQ | F1 |
| GRPO | 27.27 | 27.27 | 71.23 | 71.23 | 49.25 | 49.25 |
| DAPO | 34.68 | 34.68 | 74.10 | 74.10 | 54.39 | 54.39 |
| GRPO-UC $^{\dagger 0.5}$ | 34.21 | 18.98 | 78.89 | 71.51 | 56.55 | 45.25 |
| DAPO-UC $^{\dagger 0.5}$ | 33.55 | 22.72 | 77.65 | 75.41 | 55.60 | 49.06 |
| UCPO | 36.02 | 17.18 | 81.13 | 69.03 | 58.58 | 43.10 |
| DAPO-UCPO | 36.96 | 23.45 | 80.84 | 70.12 | 58.90 | 46.79 |

First, at the baseline level, DAPO exhibits a significant performance lead over GRPO (e.g., $54.39\%$ vs. $49.25\%$ in average PAQ), validating its superior exploration and generalization in complex reasoning. Second, when uncertainty is introduced as a simple additive reward, DAPO-UC (using the optimal coefficient of $0.5$ identified in GRPO-UC) maintains better F1 scores than GRPO-UC, particularly on GPQA-Diamond (22.72 vs. 18.98), suggesting that DAPO provides a more robust policy base for uncertainty-aware rewards. Most importantly, our full integration, DAPO-UCPO, consistently achieves the best overall performance, reaching the highest average PAQ ($58.90\%$). Notably, DAPO-UCPO effectively leverages DAPO's exploration-heavy nature to complement UCPO's calibration, resulting in a GPQA F1 score of $23.45$, which is a substantial improvement over the $17.18$ of vanilla UCPO. This synergy confirms that UCPO's advantages stem from its structural optimization rather than a simple reward penalty. By leveraging the stable gradient signals and expanded policy space provided by DAPO, UCPO achieves superior calibration and generalization, demonstrating its robust compatibility across diverse preference optimization frameworks.

### D.9. Performance Comparison Across Training Steps

This section evaluates the convergence stability and performance growth trends during model fine-tuning by comparing UCPO against GRPO-UC ($r_u = 0.8$), which represents the most competitive baseline configuration. The results across various models and tasks are illustrated in Figures 11, 12, 13, 14.

**Performance Trends and Comparative Gains.** Experimental results indicate that while UCPO's PAQ scores consistently rise and then stabilize across various tasks and models, its F1 scores remain remarkably steady. This trend stems from the model's enhanced ability to learn uncertainty representations, effectively converting previously erroneous predictions into uncertainty outputs. Notably, in the Llama-3.1-8B-Instruct math and text reasoning task, UCPO shows an initial rise in both F1 and PAQ, followed by a slight decline in the late training stage; this is likely due to the entropy collapse inherent in the GRPO algorithm, which impairs generalization. In contrast, GRPO-UC exhibits a continuous performance decline in reasoning tasks and, while showing a similar trend to UCPO in general tasks, reaches a lower performance ceiling. This disparity highlights the significant gains UCPO derives from its dynamic adjustment mechanism.

**Mechanism of Uncertainty Evolution and Metric Trade-offs.** During the training process, the proportion of uncertainty predictions is influenced by both task difficulty and model capacity, typically following an initial growth phase that leads to an eventual plateau. Correspondingly, PAQ scores improve progressively as the model learns to identify uncertainty, while the F1 score remains stable because the potential loss in recall resulting from reclassifying low-confidence yet correct predictions as uncertainty outputs is effectively offset by significant gains in precision. By generating more precise responses with fewer overconfident errors, which is the central objective of our algorithm, UCPO demonstrates superior stability and a higher performance upper bound across diverse scenarios.

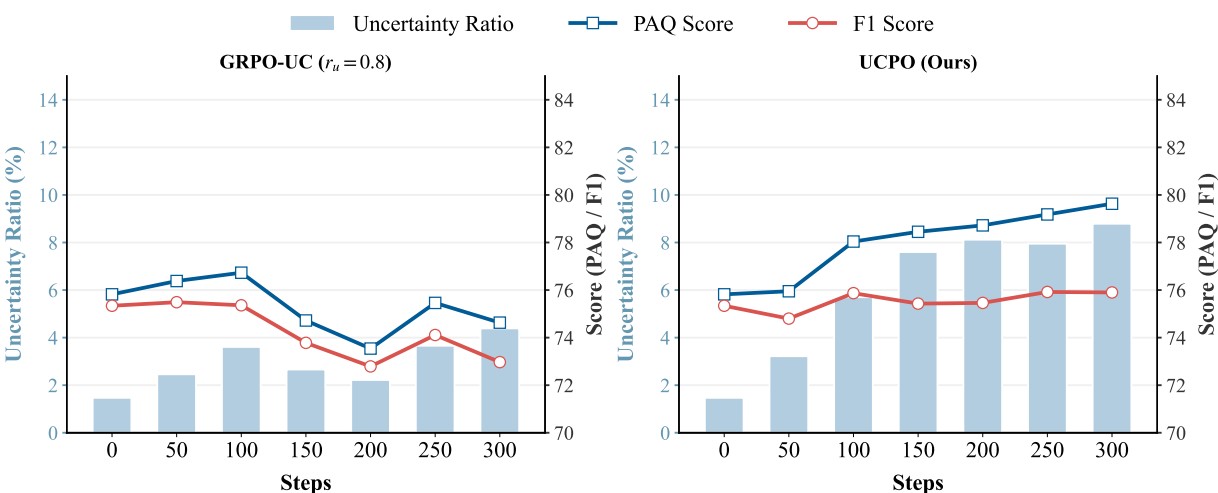

*Figure 11.* Average performance comparison between GRPO-UC and UCPO (Ours) on Qwen3-8B for Math & Text Reasoning tasks across different training steps.

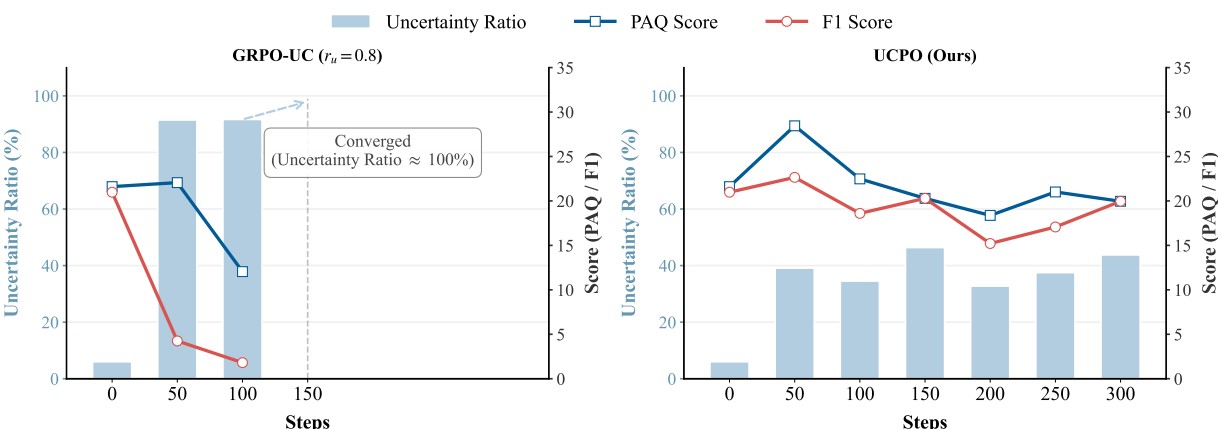

*Figure 12.* Average performance comparison between GRPO-UC and UCPO (Ours) on Llama-3.1-8B-Instruct for Math & Text Reasoning tasks across different training steps.

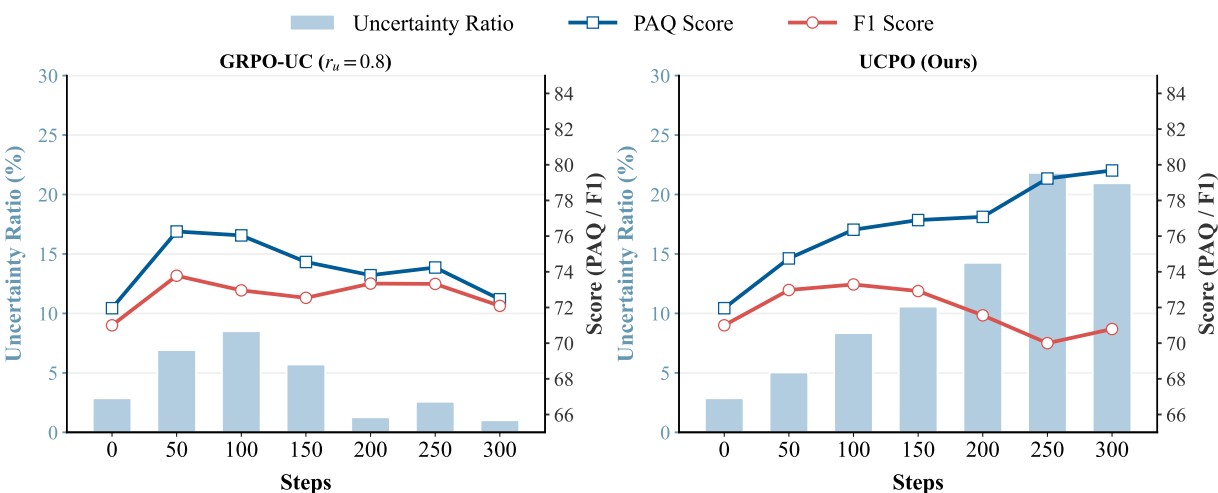

*Figure 13.* Average performance comparison between GRPO-UC and UCPO (Ours) on Qwen3-8B for General tasks across different training steps.

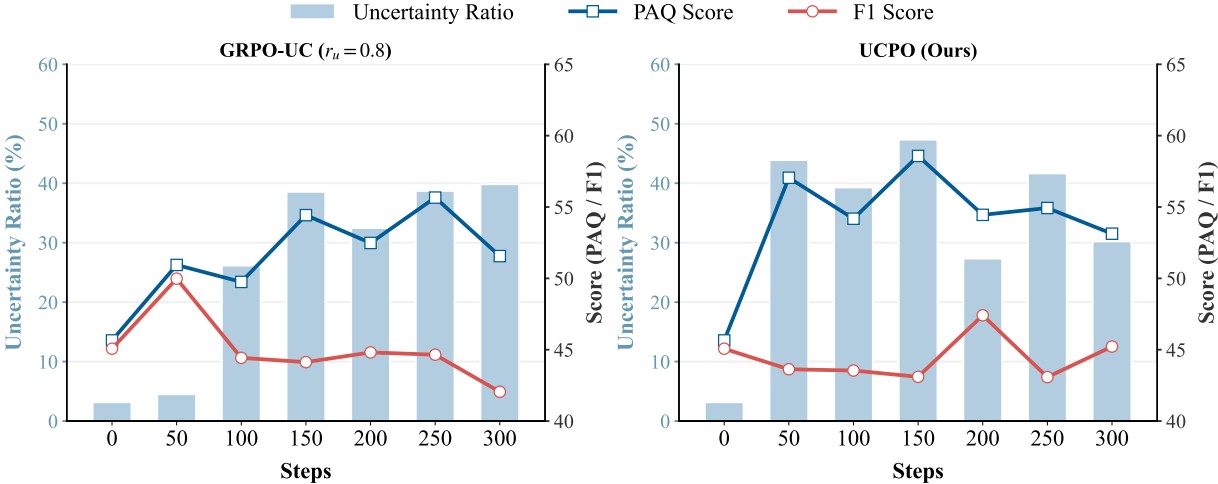

*Figure 14.* Average performance comparison between GRPO-UC and UCPO (Ours) on Llama-3.1-8B-Instruct for General tasks across different training steps.

