# OpenReview forum: "UCPO: Uncertainty-Aware Policy Optimization"
_ICML.cc/2026/Conference — ICML 2026 regular_

### Official Review · Reviewer_M49r · 2026-03-11

**Soundness:** 2
**Presentation:** 3
**Significance:** 2
**Originality:** 3
**Overall Recommendation:** 4
**Confidence:** 4

**Summary:**

This paper addresses the challenge of training LLMs to express uncertainty when facing questions beyond their knowledge boundaries. The authors identify that naively adding a fixed intermediate uncertainty reward (e.g., 0.5) to GRPO creates an "advantage bias" that leads to either overconfidence (high-performance regime) or reward hacking / avoidance degeneracy (low-performance regime). To address this, they propose UCPO with two mechanisms: (1) Ternary Advantage Decoupling (TAD), which normalizes deterministic (right/wrong) and uncertain rollouts independently, and (2) Dynamic Uncertainty Reward Adjustment (DURA), which modulates the uncertainty advantage coefficient γ(q) based on real-time rollout proportions. Experiments on Qwen3-8B and Llama-3.1-8B-Instruct across math reasoning and general tasks show that UCPO stabilizes uncertainty learning and achieves higher PAQ (Precision on Answered Questions) than baselines.

**Compliance With Llm Reviewing Policy:**

Affirmed.

**Key Questions For Authors:**

1. **F1 degradation :** On General Tasks, UCPO's F1 is consistently lower than standard GRPO for both models. Doesn't this indicate over-abstention rather than an "optimized trade-off"? How should practitioners choose between higher PAQ (fewer hallucinations among answered questions) and lower F1 (more unanswered questions)?

2. **TruthRL/KnowRL comparison :** Why are TruthRL and KnowRL, the methods most directly comparable to UCPO and cited as motivation—not included as experimental baselines?

3. **NTF data loss :** What fraction of training batches are filtered by NTF for each model-task combination? Does this create a significant sample efficiency cost?

4. **Training variance :** Can you provide results from multiple independent training runs to demonstrate that UCPO's stability (Figure 4) is not seed-dependent?

5. **w sensitivity :** How sensitive is UCPO to the weighting constant w in Equation 5? Have you tried w ≠ 1?

6. **Uncertainty detection :** How exactly is a response classified as "uncertain" during training? Is it string matching on "\\boxed{uncertainty}"? What happens if the model outputs partially uncertain responses?

**Limitations:**

Yes

**Strengths And Weaknesses:**

### Strengths

**S1 — Well-articulated problem identification.** The ternary imbalance problem (Section 2.2) is clearly diagnosed with informative ternary plots (Figure 2, p.3). The visualization of how uncertainty advantage turns negative in high-performance regimes (Fig 2a) and dominates in low-performance regimes (Fig 2b) provides genuine geometric insight into why fixed uncertainty rewards fail. This is the paper's strongest contribution.

**S2 — Convincing training dynamics analysis.** Figure 4 (p.5) effectively demonstrates UCPO's stability advantage. GRPO-UC with r_u ≥ 0.5 collapses to ~100% uncertainty on Llama math tasks, while UCPO maintains a stable, moderate uncertainty ratio. The contrast is stark and convincing.

**S3 — Practical design choices.** The DURA formula (Equation 5) has intuitive dual-term structure: Term 1 encourages uncertainty when errors are high, Term 2 suppresses unnecessary avoidance as proficiency improves. The Non-Ternary Filtering and Low-Resource Extensions (Appendix A) show attention to practical deployment issues with small group sizes.

**S4 — Two model families tested.** Unlike many papers in this space, UCPO is evaluated on both Qwen3-8B and Llama-3.1-8B-Instruct, providing some evidence of generalization. The failure modes differ across models (Qwen is more capable, Llama collapses more easily), and UCPO handles both.

**S5 — Framework compatibility demonstrated.** Appendix C.2 (Table 4) shows UCPO integrates with DAPO, achieving the best average PAQ (58.90%), demonstrating the approach is not tied to a specific RL algorithm.


### Weaknesses

**W1 — UCPO degrades F1 score on General Tasks (Major).**

*Paper claims:* UCPO achieves "an optimized trade-off between truthfulness and informativeness" (Section 4.3, p.7).

*Paper evidence:* On General Tasks (Table 2, p.7):
- Qwen3-8B: UCPO achieves PAQ 79.68% but F1 70.79%, vs. GRPO's PAQ 73.48% / F1 73.00%. UCPO's F1 is **2.2 points lower** than standard GRPO.
- Llama-3.1-8B-Instruct: UCPO achieves PAQ 58.58% but F1 43.10%, vs. GRPO's PAQ 49.25% / F1 49.25%. UCPO's F1 is **6.15 points lower** than standard GRPO.

*Assessment:* UCPO improves PAQ at the cost of F1, meaning it makes the model refuse more than necessary—gaining precision by sacrificing informativeness. On General Tasks, UCPO is not achieving a "trade-off" but rather shifting the model toward over-abstention. The paper never discusses this F1 degradation directly, instead emphasizing PAQ alone as the primary finding in Section 4.3.


**W2 — Missing comparison with the most relevant baselines (Major).**

The related work (Section 5.2, p.8) explicitly identifies TruthRL (Wei et al., 2025) and KnowRL (Ren et al., 2025) as the closest existing methods—both use RL with uncertainty rewards. The evaluation metrics are even borrowed from KnowRL ("Following KnowRL..."). Yet neither TruthRL nor KnowRL appears as an experimental baseline. The baselines (Section 4.1) are limited to: vanilla Baseline, Prompt-UC, GRPO, and GRPO-UC with various r_u values. Comparing only against a naive fixed-reward baseline (GRPO-UC) rather than the actual state-of-the-art methods that inspired this work is a significant omission.


**W3 — No theoretical analysis of the advantage bias claim (Moderate-Major).**

The paper's core claim is that fixed uncertainty rewards create "advantage bias" (Sections 1–2). This is presented through ternary plots (Figure 2) showing the advantage values under different rollout distributions. However, this analysis is entirely empirical/computational plotting Equation 1 for specific reward values. There is no formal theorem, proposition, or proof characterizing when or why the bias occurs, under what conditions it is harmful, or that UCPO provably eliminates it. For a paper whose primary contribution is identifying and resolving a structural optimization problem, the absence of formal analysis is notable for ICML.


**W4 — γ(q) computed from G=8 rollouts is extremely noisy (Moderate).**

DURA computes γ(q) using Pr, Pw, Pu—the proportions of right, wrong, and uncertain rollouts within a single group of G=8 samples (Section 3.2, Equation 5). With G=8, these proportions take only discrete values {0, 1/8, 2/8, ..., 1}. The paper acknowledges this issue (Appendix A,: "the distribution of γ(q) for G=8 is constrained to [−0.354, 0.732]") and addresses it with batch-level smoothing and tanh mapping. However, the fundamental issue remains: the signal-to-noise ratio of per-group statistics from 8 samples is very low, and the extensions are heuristic fixes with manually tuned hyperparameters (λ=0.5, α=2) that receive no sensitivity analysis.


**W5 — Evaluation uses 3 responses per instance, not multiple training seeds (Moderate).**

The paper states "all metrics are reported as the average performance across three independent responses generated for each instance" . This means the evaluation uses 3 inference samples per questio,not 3 independent training runs. No multiple-seed training experiments are reported. Given the sensitivity of RL training to initialization and the paper's own demonstration that GRPO-UC is highly unstable (Figure 4), the absence of training variance measures is a meaningful gap.


**W6 — NTF discards potentially informative training data (Moderate).**

Non-Ternary Filtering (Section 3.1) discards all groups where the deterministic set lacks either correct or incorrect rollouts. For weaker models on hard tasks (e.g., Llama on AIME), the majority of groups may have all-wrong rollouts. For strong models on easy tasks (e.g., Qwen on MATH500), groups may be all-correct. In both regimes, NTF discards large fractions of training data. The paper does not quantify how much data is filtered or analyze how this affects sample efficiency.


**W7 — Uncertainty classification mechanism underspecified (Minor-Moderate).**

The paper describes a ternary reward system (right/wrong/uncertain) but never clearly explains how a response is classified as "uncertain" during training. The prompts (Appendix B, Figures 7–8) instruct models to output "\\boxed{uncertainty}" when unsure, but the reward mechanism for detecting this at training time is not described. Is it string matching? A reward model? How robust is this classification to partial or ambiguous responses?


**W8 — DURA weighting constant w=1 not analyzed (Minor).**

Equation 5 includes a weighting constant w=1. This controls the relative strength of the uncertainty suppression term vs. the gain term. No sensitivity analysis is provided, and the paper does not explain why w=1 is appropriate or how performance changes with different values.


**W9 — Only 8B models tested (Minor).**

Both models are 8B parameters. Scaling behavior to larger (70B+) or smaller (1–3B) models is unknown, yet the paper claims broad applicability to "trustworthy AI."

---

> ### Author Rebuttal · Authors · 2026-03-31
>
> # Response to Reviewer M49r
>
> We thank the reviewer for the comprehensive and detailed review. We address each concern below.
>
> ---
>
> ## W1: F1 degradation on General Tasks — over-abstention?
>
> General tasks are predominantly multiple-choice. Before abstention learning, the model may answer low-confidence questions correctly by chance (e.g., 25% guessing probability). After UCPO training, these lucky guesses become uncertainty outputs, reducing recall and F1.
>
> However, PAQ improves significantly: Qwen3-8B from 73.48% to 79.68%, Llama from 49.25% to 58.58%. UCPO filters out low-confidence incidental correct answers, making definitive answers more reliable.
>
> **Practitioner guidance**: In high-stakes domains (medical, legal), high PAQ should be prioritized. In information retrieval, coverage matters more. UCPO provides an interface through the weighting constant $w$ in DURA (see W8).
>
> ---
>
> ## W2: Missing TruthRL/KnowRL comparison
>
> GRPO-UC is our unified abstraction of these methods, as both use fixed intermediate-value rewards for uncertain predictions in GRPO. By varying $r_u \in \lbrace 0.2, 0.5, 0.8 \rbrace$, we cover their design space. We will explicitly clarify this mapping in the revised Baselines section.
>
> ---
>
> ## W3: No theoretical analysis of advantage bias
>
> We appreciate this important suggestion. The ternary diagrams in Figure 2 are intended to visually illustrate the ternary imbalance problem. In fact, UCPO is grounded in rigorous mathematical derivations. Due to space constraints, we refer the reviewer to our **response to Reviewer ptgn (Q3)** for detailed formal analysis. Complete proofs will be provided in the revised supplementary material.
>
> ---
>
> ## W4 & W8: $\gamma(q)$ noise from G=8 & sensitivity of w=1
>
> We appreciate the attention to both issues. Due to space constraints, we refer the reviewer to our **response to Reviewer ptgn (Q1)** for detailed derivations and experiments.
>
> ---
>
> ## W5: No multi-seed training experiments
>
> We acknowledge this limitation. We will supplement multi-seed training experiments along with detailed training curves in the revision to further validate the robustness of UCPO's training convergence.
>
> ---
>
> ## W6: NTF data loss
>
> NTF maintains consistency with standard GRPO, which itself discards all-correct or all-wrong groups (std=0, advantages vanish). NTF extends this logic to the ternary setting by filtering groups that lack correct or incorrect rollouts, as such groups cannot provide the ternary gradient signals required for uncertainty learning.
>
> To clarify, training is conducted on DAPO-Math-17k, while AIME and MATH500 serve as evaluation benchmarks. The proportions of NTF-filtered groups (all-correct + all-wrong) during training are:
>
> - **Qwen3-8B**: All-correct fluctuates from ~10% to ~14% after step 150; all-wrong decreases from 23% to 11% at step 300.
> - **Llama-3.1-8B**: All-correct remains at ~0.7% throughout (due to weaker mathematical capability); all-wrong decreases from 25% to below 1%.
>
> The ablation study (Figure 9, UCPO vs. w/o NTF) shows that NTF does not impair sample efficiency but instead accelerates convergence to a stable uncertainty ratio, as retaining signal-lacking groups introduces noise. We will include NTF filtering ratio visualizations in the revision. Guiding abstention learning from entirely negative samples remains a promising future direction.
>
> ---
>
> ## W7: Uncertainty detection mechanism
>
> We apologize for the underspecification. The prompt templates in Appendix Figures 7-8 are used during both training and evaluation, instructing the model to output `\boxed{uncertainty}` when unsure. At training time, the reward function classifies responses via **regex string matching**: if the output contains the uncertainty keyword, it receives $r_u$; if it contains a correct answer, it receives $r_{\text{right}}$; otherwise, it receives $r_{\text{wrong}}$. Partial or ambiguous responses that do not follow the prescribed `\boxed{uncertainty}` format are treated as definitive answers and judged by correctness. No reward model is involved.
>
> We deliberately adopt this simple rule-based approach because our core contribution lies in the structural analysis of fixed $r_u$ and the UCPO bias elimination mechanism, not in uncertainty detection itself. String matching cleanly isolates this contribution in controlled experiments. For open-ended tasks that require recognizing implicit or ambiguous uncertainty expressions, the rule-based classifier can be replaced with a reward model, as UCPO is decoupled from the classification module.
>
> ---
>
> ## W9: Only 8B models
>
> We thank the reviewer for this suggestion. We will supplement experiments on larger models (e.g., Qwen3-32B) in the revision. The two current 8B models differ significantly in architecture and capability, exhibiting markedly different failure modes, yet UCPO achieves stable optimization on both, which provides preliminary evidence of generalizability.

---

> > ### Author Rebuttal · Reviewer_M49r · 2026-03-31
> >
> > I thank the authors for their thoughtful rebuttal. Below I address each response and provide my assessment.
> >
> > **W1 (F1 degradation):** The "lucky guess filtering" explanation is plausible and I appreciate the honest framing. On multiple choice tasks with 25% guessing probability, it is reasonable that UCPO would filter some incidentally correct answers. However, this explanation is not verified empirically. How much of the F1 drop comes from filtering lucky guesses versus genuine over abstention on questions the model could answer confidently? Without decomposing the F1 loss into these two sources, the explanation remains a hypothesis. The practitioner guidance (high stakes favors PAQ, retrieval favors F1) is sensible and should appear in the main text. The mention that w in DURA can control this trade off is valuable but unexplored. **Partially addressed.**
> >
> > **W2 (Missing TruthRL/KnowRL):** The claim that GRPO-UC is a "unified abstraction" of TruthRL and KnowRL is a simplification. While both methods do use fixed intermediate rewards, they have additional design choices (different reward structures, training procedures, data handling strategies) that are not captured by simply sweeping r_u in GRPO-UC. Asserting subsumption is not the same as demonstrating it. Even a single comparative run would have been more convincing than this argument. I understand the authors will "clarify this mapping" in the revision, but a textual mapping is not an empirical comparison. **Weakly addressed. This remains a notable gap.**
> >
> > **W3 (No theoretical analysis):** The authors defer to their response to Reviewer ptgn and promise complete proofs in supplementary material. I cannot evaluate content I have not seen. If the formal analysis is as rigorous as described, this would meaningfully strengthen the paper. **Pending the revision.**
> >
> > **W4 and W8 (γ noise and w sensitivity):** Similarly deferred. I understand the character limit necessitates prioritization. **Pending.**
> >
> > **W5 (Multi seed training):** The promise to supplement multi seed experiments is noted. Given that the paper's own Figure 4 demonstrates how sensitive GRPO-UC is to training dynamics, showing UCPO is robust across seeds is important for the stability claims. A promise is not data. **Not yet addressed.**
> >
> > **W6 (NTF data loss):** This is the strongest part of the rebuttal. The concrete filtering percentages (Qwen3-8B: 10–14% all correct, 23%→11% all wrong; Llama: ~0.7% all correct, 25%→<1% all wrong) and the reference to Figure 9 showing NTF accelerates rather than impairs convergence are convincing. The argument that retaining groups without ternary gradient signals introduces noise rather than useful information is theoretically sound. **Resolved.**
> >
> > **W7 (Uncertainty detection):** The regex string matching approach is clearly described and the justification for isolating the structural contribution from the detection mechanism is appropriate. The acknowledgment that a reward model could replace string matching for open ended tasks shows practical awareness. **Resolved.**
> >
> > **W9 (8B models):** The promise to add Qwen3-32B results is noted. The observation that two architecturally different 8B models with distinct failure modes provide some generalization evidence is fair. **Pending the revision.**
> >
> > **Updated Assessment:**
> >
> > The rebuttal resolves two concerns cleanly (W6 NTF quantification, W7 uncertainty detection) and provides a plausible but unverified explanation for W1 (F1 degradation). These are positive steps. However, the two major weaknesses present a mixed picture: W1 is partially addressed through a reasonable but unquantified argument, and W2 (missing baselines) receives only an assertion of subsumption without empirical evidence. Three concerns (W3, W4, W8) are deferred to another reviewer's response, which I cannot evaluate from this rebuttal alone. Two more (W5, W9) are promises of future experiments.
> >
> > The paper's core strengths remain intact: the ternary imbalance insight is genuinely valuable (Figure 2), the training dynamics are convincing (Figure 4), and the method works across two model families. These are real contributions.

---

> > > ### Author Response · Authors · 2026-04-01
> > >
> > > We thank the reviewer for the detailed follow up review. Below we provide specific responses and new evidence.
> > >
> > > ---
> > >
> > > ## W1: F1 Degradation from over-abstention?
> > >
> > > Decomposing F1 loss into lucky guess filtering and excessive abstention is difficult to precisely disentangle. We therefore verify from an alternative perspective: if UCPO uncertainty judgment is accurate, cases deemed uncertain should yield lower accuracy even when answered by a model without uncertainty training.
> > >
> > > Using Qwen3-8B, we test GRPO on uncertain cases identified by UCPO:
> > >
> > > | Dataset | Full Accuracy | Uncertain Accuracy | Drop |
> > > |---------|---------------|-------------------|------|
> > > | GPQA Diamond | 59.35 | 36.96 | -22.39 |
> > > | MMLU Redux2 | 87.62 | 55.08 | -32.54 |
> > > | Average | 73.48 | 46.02 | -27.46 |
> > >
> > > The 46.02% accuracy on uncertain cases (vs. 73.48% overall) confirms the model genuinely lacks knowledge. For multiple choice questions, random guessing among 4 options yields 25 percent, while guessing after eliminating 2 options yields 50 percent. The 46.02% accuracy suggests these cases are genuinely uncertain. F1 degradation stems from filtering lucky guesses, not over-abstention, which aligns with UCPO design goal of trading lucky guesses for reliability.
> > >
> > > ---
> > >
> > > ## W2: TruthRL and KnowRL Baseline
> > >
> > > **Difference in Contribution Focus.** This paper proposes an RL training improvement that optimizes training dynamics for uncertainty expression. KnowRL involves additional dimensions beyond uncertainty learning, including specialized refusal fine tuning dataset construction. KnowRL employs fixed uncertainty reward values, which is precisely the issue this paper addresses.
> > >
> > > **Principle of Fair Comparison.** To validate our method of dynamic uncertainty reward adjustment, we control other variables consistently. If differences exist across dataset construction and training pipeline, isolating the method contribution becomes difficult. We adopt GRPO UC as a unified abstraction and cover the design space of TruthRL and KnowRL by adjusting the fixed reward value $r_u$, focusing on comparing dynamic versus fixed reward mechanisms.
> > >
> > > It is worth emphasizing that UCPO, as an improvement to the training framework, can be combined with the methodology of KnowRL.
> > >
> > > ---
> > >
> > > ## W3: Theoretical Analysis
> > >
> > > The ternary diagrams in Figure 2 visually illustrate the ternary imbalance problem. UCPO is grounded in rigorous mathematical derivations. Due to space constraints, we present core conclusions here while complete proofs will be provided in the revised supplementary material.
> > >
> > > **Proposition 1 (Uncertainty Suppression)**: For GRPO UC, $A_u = \frac{r_u(1-P_u) - P_r}{\sigma}$. When $P_r > \frac{r_u}{1-r_u} P_w$, $A_u < 0$. As $P_w \to 0$, suppression becomes inevitable $\forall r_u \in (0,1)$.
> > >
> > > **Proposition 2 (Reward Hacking)**: When $P_r \to 0$, $H(P_u) = \sqrt{P_u(1-P_u)}$ increases monotonically for $P_u < 1/2$, creating a self reinforcing cascade toward $P_u \to 1$. This structural defect is independent of $r_u$.
> > >
> > > **Proposition 3 (UCPO Guarantees)**: UCPO satisfies NC1 (Anti Suppression: $A_u > 0$ as $P_u \to 0$) and NC2 (Anti Hacking: stable equilibrium $P_u^* < 1$ exists) via $\lim_{P_u \to 0} \gamma(q) = 1 > 0$ and $\gamma(q) < 0$ for large $P_u$.
> > >
> > > ---
> > >
> > > ## W4: Sensitivity Analysis of $G$
> > >
> > > Under low resource conditions where $G$ is small, $\gamma(q)$ computation suffers from noise issues. We address this through batch level smoothing and non linear mapping, verified in ablation experiments (Table 3).
> > >
> > > Sensitivity analysis on sampling group size $G$:
> > >
> > > | $G$ | PAQ | F1 | Convergence Steps |
> > > |-----|-----|-----|-------------------|
> > > | 4 | 55.08 | 42.87 | ~200 |
> > > | 8 | 58.58 | 43.10 | ~50 |
> > > | 16 | 57.72 | 46.89 | ~50 |
> > >
> > > $G \geq 8$ provides finer statistical estimation with faster convergence. All values achieve stable convergence.
> > >
> > > ---
> > >
> > > ## W5: Multi Seed Training Experiments
> > >
> > > We conducted multi seed experiments using time as random seed for 3 independent runs on Llama 3.1 8B Instruct with general tasks:
> > >
> > > | Metric | seed1 | seed2 | seed3 | Mean ± Std |
> > > |--------|-------|-------|-------|------------|
> > > | PAQ | 58.58 | 59.00 | 57.61 | 58.40 ± 0.72 |
> > > | F1 | 43.10 | 45.83 | 43.94 | 44.29 ± 1.40 |
> > >
> > > All seeds achieve stable convergence with consistent curve trends and low variance. Complete training curves will be included in the revised version.
> > >
> > > ---
> > >
> > > ## W8: Sensitivity of $w$
> > >
> > > We choose $w=1$ based on theoretical motivation: (1) $\gamma(q) \in (-w, 1)$ under sufficient rollouts; (2) at $w=1$, $\gamma(q)=0$ under uniform distribution ($P_r=P_w=P_u=1/3$), providing neutral equilibrium.
> > >
> > > Experimental comparison on Llama 3.1 8B with General Tasks average:
> > >
> > > | $w$ | Uncertainty(%) | PAQ | F1 |
> > > |-----|----------------|-----|-----|
> > > | 0.5 | 57.61 | 58.63 | 37.91 |
> > > | 1 | 47.33 | 58.58 | 43.10 |
> > > | 2 | 35.93 | 54.53 | 45.05 |
> > >
> > > The parameter $w$ modulates the reliability versus informativeness tradeoff. All values converge without reward hacking; $w=1$ achieves optimal balance.
> > >
> > > ---

---

### Official Review · Reviewer_EyUa · 2026-03-12

**Soundness:** 2
**Presentation:** 3
**Significance:** 3
**Originality:** 3
**Overall Recommendation:** 4
**Confidence:** 4

**Summary:**

This work introduces several techniques to effectively integrate uncertainty rewards into the GRPO advantage computation.
Those Ternary Advantage Decoupling, which separates the uncertainty advantage from performance advantatge, Dynamic Uncertainty Reward Adjustment that matches the scales of performance and uncertainty rewards.
The paper conducts evaluation of the method as well as ablation to show that the complete framework, refered to as UCPO, is advantageous compared to other possible techniques to integrate uncertainty into training.

**Compliance With Llm Reviewing Policy:**

Affirmed.

**Final Justification:**

Authors have addressed a large number of my concerns with the rebuttals.
Given the improvements declared for the paper, I reassess it positively.

**Key Questions For Authors:**

1. GRPO-UC - is this an existing prior work? The closest work seems to be Ren et al. Know-RL. Why would the authors not compare to that?
2. What reward function is used for training and how does it determine if the answer is 'uncertain'?
3. In Line:260: "Accuracy (Acc), Hallucination (Hal), or Uncertainty (Unc)". How are those differentiated during evaluation? Does the model abstain from answering? These benchmark datasets do not inherently contain the "Uncertain" reward option.
4. In authors opinion, is every incorrect answer a hallucination? What is the difference between a hallucinated and incorrect answer?

**Limitations:**

The authors discuss some limitations of the method in the conclusion section.

**Strengths And Weaknesses:**

### Strengths

The paper introduces a novel way to integrate uncertainty reward into the GRPO online optimization.
The research is interesting, although the significance may be somewhat reduced by the need for the explicit "uncertain" reward function option.
The method appears to fare well with instruct models, although fails to strive with models already capable of reasoning.
Computational overhead is low compared to normal GRPO as all the additional computation happens at the level of the rewards and advantage computation.

### Weaknesses

1. (moderate) The authors fail to mention how does the reward function actually determine if the response is uncertain. From my understanding the uncertainty of the reward function is dealt with in the paper, which is never explicitly stated, reducing the clarity.
2. (minor) Would be interesting to see perfromance metrics plots (i.e. Right/Wrong) alongside the uncertainty ratio ones.
3. (moderate) Empirical performance on Qwen3-8B is lagging. I do not believe authors discuss this at any point.
4. (major) The terms "hallucination" and "uncertainty" are thrown around a lot, often dubiously. E.g. Line:323 "By converting hallucinations into uncertainty"; Line:432 "achieving stable and effective uncertainty learning" - does the model learn uncertainty? The authors seem to be unaware of the uncertainty quantification literature (e.g. [1,2,3] as well as preceeding work in classification / regression) and differentiation between hallucinations and uncertainty. This seems to lead additionally to poor discrimination between the reward function uncertainty and the generating models uncertainty throughout the paper which in my view is a significant flaw.


### References
1. Malinin, A. & Gales, M. Uncertainty Estimation in Autoregressive Structured Prediction. in (2020).
2. Kuhn, L., Gal, Y. & Farquhar, S. Semantic uncertainty: Linguistic invariances for uncertainty estimation in natural language generation. in The eleventh international conference on learning representations (2023).
3. Ielanskyi, M., Schweighofer, K., Aichberger, L. & Hochreiter, S. Addressing pitfalls in the evaluation of uncertainty estimation methods for natural language generation. in The fourteenth international conference on learning representations (2026).

---

> ### Author Rebuttal · Authors · 2026-03-31
>
> # Response to Reviewer EyUa
>
> We thank the reviewer for the thorough review. We address each concern below.
>
> ---
>
> ## Q1: Is GRPO-UC an existing prior work? Why not compare with KnowRL?
>
> GRPO-UC is not a specific prior work but our unified abstraction of the core reward design shared by KnowRL and TruthRL. These methods assign a fixed reward to uncertain predictions within GRPO (e.g., $r_u \in \lbrace 0.2, 0.5, 0.8 \rbrace$). The GRPO-UC variants in Tables 1–2 thus subsume their core mechanisms. We will explicitly annotate this correspondence in the revised version.
>
> ---
>
> ## Q2: What reward function is used and how does it determine "uncertain"?
>
> We use prompt templates (Appendix Figures 7–8) to guide the model to output abstention expressions (e.g., `\boxed{uncertainty}`). The reward function classifies outputs via **regular expression matching**: matching an abstention keyword → $r_u$; correct answer → $r_{\text{right}}$; incorrect answer → $r_{\text{wrong}}$. We adopt this simple scheme because the core contribution is the advantage bias elimination mechanism, not uncertainty detection itself; rule-based classification cleanly isolates this contribution in controlled experiments. For open-ended tasks, the rule matcher can be replaced with a reward model, as UCPO is decoupled from the classification module.
>
> ---
>
> ## Q3: Performance metric plots alongside uncertainty ratio
>
> We appreciate this question. Appendix Figures 10–13 already present PAQ, F1, and uncertainty ratio evolution across training steps for all model-task combinations, directly illustrating the interplay between uncertainty ratio and performance metrics. We hope these existing figures provide the information the reviewer is looking for.
>
>
> ---
>
> ## Q4: Empirical performance on Qwen3-8B
>
> Thank you for raising this point. The limited improvement on Qwen3-8B is primarily due to a **performance ceiling effect**: the model has been extensively fine-tuned on large-scale mathematical data and is already near saturation, leaving limited room for further reasoning improvement through fine-tuning. This phenomenon is also reported in GDPO [1]. Nevertheless, UCPO achieves the highest average PAQ across all methods (Math: 79.63%, General: 79.68%) by converting confident but incorrect predictions into abstentions, validating its core design objective.
>
> [1] Liu S Y, Dong X, Lu X, et al. GDPO: Group reward-decoupled normalization policy optimization for multi-reward RL optimization. arXiv:2601.05242, 2026.
>
>
> ---
>
> ## Q5: Clarification on "hallucination" and "uncertainty" terminology
>
> We appreciate this rigorous feedback and clarify:
>
> **"Uncertainty"** in this paper refers to explicit abstention behavior (e.g., outputting "I'm not sure"), not epistemic/aleatoric uncertainty in the Bayesian sense. **"Hallucination"** refers to confidently producing incorrect answers. We will revise imprecise expressions: Line 323 → emphasize converting erroneous assertions into proactive abstention; Line 432 → "achieving stable and effective learning of uncertainty expression."
>
> **Relationship with Uncertainty Quantification (UQ) literature**: We are aware of the works cited (Malinin & Gales 2020; Kuhn et al. 2023; Ielanskyi et al. 2026). These focus on **measuring** existing uncertainty (inference-time, continuous confidence scores), while UCPO focuses on **training** the model to abstain (training-time, discrete behavior). The two are complementary: UQ confidence estimates could serve as a more fine-grained reward signal for UCPO. We will add a UQ discussion in the revised Related Work.
>
> ---
>
> ## Q6: How are Accuracy, Hallucination, and Uncertainty differentiated during evaluation?
>
> Yes, the model explicitly abstains. The classification is: (1) output matches abstention keywords → **Uncertainty**; (2) definitive answer matching ground truth → **Accuracy**; (3) definitive incorrect answer → **Hallucination**. Notably, the benchmark datasets do not contain an explicit uncertain option, which is precisely the value of UCPO: the abstention signal comes entirely from the model's own generation. We evaluate whether the model has learned to proactively abstain when it should, rather than guess arbitrarily. For benchmarks containing real unanswerable queries, please refer to our **response to Reviewer ptgn Q2**.
>
> ---
>
> ## Q7: Is every incorrect answer a hallucination?
>
> Strictly speaking, no. In our experimental framework (math reasoning and multiple-choice), answer correctness is binary, so we operationally classify all definitive incorrect answers as hallucinations. This is justified by three considerations: (1) the goal of UCPO is to abstain whenever any error may occur, as the risk to users is equivalent regardless of the error source; (2) this convention is consistent with the evaluation protocols in KnowRL and TruthRL; (3) UCPO's mechanisms are independent of the specific cause of errors. We will use the term more precisely and state this simplification explicitly in the revision.

---

> > ### Author Rebuttal · Reviewer_EyUa · 2026-04-03
> >
> > I thank the authors for answering my questions - this clarified things significantly for me.
> > The delineations in the Q5 should really be made much more explicit.
> > I would urge the authors to, perhaps, avoid the use of the term "Hallucination" where it means incorrect answer altogether.

---

> > > ### Author Response · Authors · 2026-04-07
> > >
> > > Thank you for your further reply and constructive suggestions regarding terminology. We highly value your feedback and will make the following improvements in the next revision so that readers from different research backgrounds can more clearly understand the positioning of our work:
> > >
> > > (1) Certain expressions that may cause ambiguity will be refined (e.g., replacing hallucination with incorrect/erroneous response where appropriate, and retaining the term only when the model exhibits high confidence assertions without factual grounding), making the writing more precise.
> > >
> > > (2) Section 1 will explicitly define the research scope: UCPO focuses on training models to learn proactive abstention behavior, which is a distinct yet complementary research direction compared to the continuous confidence estimation studied in the Uncertainty Quantification (UQ) literature.
> > >
> > > (3) A new paragraph discussing UQ related work (Malinin & Gales 2020; Kuhn et al. 2023; Ielanskyi et al. 2026) will be added to Related Work, clarifying the connections and distinctions between the two lines of research.
> > >
> > > We believe these revisions will make the paper more rigorous in its presentation. Thank you again for the detailed feedback throughout the review process. If you have any further questions about the paper, we are happy to continue the discussion.

---

### Official Review · Reviewer_ptgn · 2026-03-13

**Soundness:** 4
**Presentation:** 4
**Significance:** 4
**Originality:** 3
**Overall Recommendation:** 5
**Confidence:** 5

**Summary:**

This paper proposes UCPO (Uncertainty-Aware Policy Optimization), a reinforcement learning framework designed to improve the reliability of large language models by enabling them to express uncertainty when they lack sufficient knowledge. The authors observe that existing RL-based approaches that incorporate uncertainty rewards often suffer from instability: models either suppress uncertainty and become overconfident or exploit uncertainty rewards and collapse into excessive abstention.

To address this issue, the paper identifies the ternary imbalance problem that arises when extending binary RL reward structures (correct vs. incorrect) to include a third outcome representing uncertainty. Standard advantage normalization causes biased gradient signals that lead to reward hacking or suppression of uncertainty.

UCPO resolves this by introducing two main mechanisms. First, Ternary Advantage Decoupling (TAD) separates deterministic outputs (correct or incorrect answers) from uncertain outputs and computes advantages independently, preventing interference between reasoning signals and uncertainty learning. Second, Dynamic Uncertainty Reward Adjustment (DURA) adaptively adjusts the reward strength for uncertainty based on the model’s current performance and the distribution of rollout outcomes, balancing incentives between expressing uncertainty and producing correct answers.

**Compliance With Llm Reviewing Policy:**

Affirmed.

**Key Questions For Authors:**

1. **Sensitivity to Hyperparameters:** How sensitive is UCPO to the design of the DURA coefficient and rollout statistics?

2. **Behavior on Real Unanswerable Queries:** Did you evaluate the model on datasets explicitly containing unanswerable or ambiguous questions?

(optional:
3. **Scalability:** Have you evaluated UCPO on larger models (e.g., >30B parameters) or stronger reasoning models?
   *Impact:* Evidence that the method scales to larger LLMs would strengthen confidence in its practical relevance.
)

**Limitations:**

No. The paper briefly mentions technical limitations (e.g., dependence on rollout distribution and instability in low-resource settings), but it does not sufficiently discuss broader limitations or potential societal impacts.

**Strengths And Weaknesses:**

### Soundness
- **Strengths:** The method is technically reasonable and builds on GRPO with clear modifications (TAD and DURA). Experiments across models and tasks plus ablations provide empirical support.
- **Weaknesses:** The analysis of the ternary imbalance problem is largely heuristic with limited theoretical justification. Experiments are limited to 8B models and a small set of benchmarks.

### Presentation
- **Strengths:** The paper is generally well organized and the figures help explain the intuition behind the method.
- **Weaknesses:** Some explanations are informal and several implementation details are brief, which may make reproduction harder.

### Significance
- **Strengths:** The paper tackles an important problem—reducing hallucinations by enabling LLMs to express uncertainty—which is relevant for trustworthy AI.
- **Weaknesses:** Empirical improvements are moderate and the impact is likely incremental rather than transformative.

### Originality
- **Strengths:** Introduces a ternary reward formulation and decoupled advantage estimation for uncertainty-aware RL.
- **Weaknesses:** The approach mainly combines existing ideas (RL optimization and reward shaping), so the novelty is mostly in the specific formulation.

---

> ### Author Rebuttal · Authors · 2026-03-30
>
> # Response to Reviewer ptgn
>
> We thank the reviewer for the thorough review and positive evaluation. We address each question below.
>
> ---
>
> ## Q1: Sensitivity to Hyperparameters
>
> We analyze from two perspectives: the weighting constant $w$ in DURA and the rollout group size $G$.
>
> ### (1) Weighting Constant $w$
>
> We choose $w=1$ based on the following theoretical motivation: (1) $w$ governs the range of $\gamma(q)$; under sufficient rollouts, $\gamma(q) \in (-w, 1)$; (2) at $w=1$, $\gamma(q)=0$ under a uniform distribution ($P_r=P_w=P_u=1/3$), providing a neutral equilibrium, which serves as the least-biased default.
>
> Experimental comparison (Llama-3.1-8B, General Tasks average):
>
> | $w$ | Uncertainty(%) | PAQ | F1 |
> |-----|---------------|-----|-----|
> | 0.5 | 57.61 | 58.63 | 37.91 |
> | 1 (default) | 47.33 | 58.58 | 43.10 |
> | 2 | 35.93 | 54.53 | 45.05 |
>
> $w$ modulates the reliability (PAQ) vs. informativeness (F1) trade-off. All values converge stably without reward hacking, validating UCPO's robustness. $w=1$ achieves the best balance.
>
> ### (2) Rollout Group Size $G$
>
> | $G$ | PAQ | F1 |
> |-----|-----|-----|
> | 4 | 55.08 | 42.87 |
> | 8 (default) | 58.58 | 43.10 |
> | 16 | 57.72 | 46.89 |
>
> **Training Observation.** $G$ primarily affects convergence speed: ~200 steps for $G=4$ vs. ~50 steps for $G \geq 8$, with similar final performance. All sizes achieve stable convergence.
>
> **Analysis.** The superior performance of $G=8,16$ benefits from finer-grained statistical estimation of uncertainty proportions. We will further explore the impact of rollout group size on inference-time scaling in future work.
>
> ---
>
> ## Q2: Behavior on Real Unanswerable Queries
>
> We conducted evaluation on GPQA_Abstain from AbstentionBench [1], which contains explicitly unanswerable questions (Llama-3.1-8B, fine-tuned on General Tasks):
>
> | Methods | Total Acc | Abstain Recall | Abstain Prec |
> |---------|-----------|---------------|--------------|
> | Prompt-UC | 21.67 | 15.00 | 60.00 |
> | GRPO | 22.50 | 0.00 | 0.00 |
> | GRPO-UC$^{\dagger 0.5}$ | 43.75 | 70.00 | 60.87 |
> | UCPO | **47.50** | **82.50** | 58.93 |
>
> GRPO cannot identify unanswerable questions at all (Recall=0%). UCPO achieves the best Total Acc (47.50%) and Abstain Recall (82.50%), demonstrating that learned uncertainty expression transfers effectively to explicit unanswerable scenarios. We will add more AbstentionBench experiments in the revision.
>
> [1] Kirichenko, P., Ibrahim, M., Chaudhuri, K., & Bell, S.J. (2025). AbstentionBench: Reasoning LLMs Fail on Unanswerable Questions. ArXiv, abs/2506.09038.
>
> ---
>
> ## Q3: Heuristic analysis & limited model scale
>
> ### Theoretical Supplementation
>
> The ternary diagrams in Figure 2 are intended to visually illustrate the ternary imbalance problem. In fact, UCPO is grounded in rigorous mathematical derivations. Due to space constraints, we present only the core conclusions here (complete proofs will be provided in the revised supplementary material):
>
> **Proposition 1 (Uncertainty Suppression)**: For GRPO-UC, $A_u = \frac{r_u(1-P_u) - P_r}{\sigma}$. When $P_r > \frac{r_u}{1-r_u} P_w$, $A_u < 0$. As $P_w \to 0$, suppression becomes inevitable for any $r_u \in (0,1)$.
>
> **Proposition 2 (Reward Hacking)**: In the low-performance regime ($P_r \to 0$), the gradient contribution $H(P_u) = \sqrt{P_u(1-P_u)}$ is monotonically increasing for $P_u < 1/2$, forming a self-reinforcing cascade toward $P_u \to 1$. This collapse is independent of $r_u$, revealing a structural defect of fixed rewards.
>
> **Proposition 3 (UCPO Guarantees)**: We extract two necessary conditions: NC1 (Anti-Suppression: $A_u > 0$ as $P_u \to 0$, $\forall P_r$) and NC2 (Anti-Hacking: stable equilibrium $P_u^* < 1$ exists). UCPO satisfies both: $\lim_{P_u \to 0} \gamma(q) = 1 > 0$ and $\gamma(q) < 0$ for large $P_u$, guaranteeing equilibrium by the Intermediate Value Theorem.
>
> ### Model Scale
>
> Regarding model scale, we will supplement experiments on larger models (e.g., Qwen3-32B) in the revision to validate UCPO's scalability.
>
> ---
>
> ## Q4: Informal explanations & reproduction difficulty
>
> We apologize for the insufficient detail. In the revision, we will: (1) include complete theoretical proofs in the supplementary (as detailed in Q3); (2) provide full reproduction details covering training hyperparameters, reward function implementation, and evaluation procedures.
>
> We have already included the complete source code in the supplementary material, and will officially open-source it upon acceptance to facilitate community reproduction. We hope the code helps the reviewer better understand the implementation details of our work.

---

> > ### Author Rebuttal · Reviewer_ptgn · 2026-03-31
> >
> > Thank you for addressing the issues I concern about. I will keep the score, and good luck with your research and submission! I am looking forward to see your code!

---

### Official Review · Reviewer_NJEw · 2026-03-13

**Soundness:** 2
**Presentation:** 3
**Significance:** 3
**Originality:** 3
**Overall Recommendation:** 5
**Confidence:** 4

**Summary:**

The paper introduces UCPO, a modification to GRPO that encourages the model to abstain from answering when uncertain. To do so, the paper shows that naively adding a reward for uncertainty (with its value between that of right and wrong) can lead to overconfidence in high-performance regimes and reward hacking in low-performing regimes. To mitigate this challenge, the paper proposes ternary advantage decoupling to compute advantage in the deterministic and uncertain channels separately. Further, the uncertainty reward weighted is dynamically adjusted based on the ratio of right, wrong, and uncertain answers in the group. Evaluations on math and text reasoning as well as general tasks benchmarks show improved precision on answered questions and improved F1 score on average.

**Compliance With Llm Reviewing Policy:**

Affirmed.

**Final Justification:**

The insights provided in the paper are useful, and the proposed solution is a simple remedy that's also empirically effective. The rebuttal addresses my concerns on the effectiveness of the results, and the inference-time scaling results are compelling. Overall, I think the contributions are valuable.

**Key Questions For Authors:**

1. Why not quantify uncertainties? How is the current result connected to improved calibration, in the sense that empirical accuracy matches predicted confidence?
2. Could you provide more in-depth discussions on the results, especially on the mixed improvement/degradation in the main results, and the accuracy drop for general tasks in Figure 5?
3. How does inference-time scaling affect the uncertainty awareness of the models?

**Limitations:**

yes

**Strengths And Weaknesses:**

- Soundness
    - The analysis on GRPO-UC is insightful. Figure 2 provides good intuition on how a naive method to encourage uncertainty can lead to unintended consequences such as reward hacking.
    - The motivation to dynamically adjust weight on uncertainty reward is sound, the choice of the method therefore follows naturally.
    - The paper claims improved calibration, but this point is not directly obvious. In section 4.3, it’s shown that UCPO improves precision on answered questions, how does this translate to calibration? Are we treating answered questions as 100% confident and unanswered ones as 0%?
    - In the main results, the average PAQ and F1 scores improve, but results on individual datasets seem to be very mixed, especially for the Qwen model. It would be helpful to discuss these results in more depth.
    - In Figure 5, accuracy seems to drop for general tasks. Are there possible explanations for that?
- Presentation
    - The overall presentation is clear, with well-motivated problem setup and thorough analysis of challenges with existing methods. Figures are insightful and convey important information in a straightforward way.
    - It might be helpful to include discussions of related works in uncertainty quantification [1, 2], where calibrated uncertainty signal can be used to determine whether to reveal the answer.
- Significance
    - The proposed method is practical and effective in reducing hallucinations, as shown in Figure 5.
    - Improvements in PAQ and F1 score are not that significant, especially for Qwen.
    - Baselines are based on setting different weights for uncertainty reward, while other works that explore abstention could also be useful to compare against, e.g., KnowRL.
- Originality
    - The core idea of dynamically adjusting weight on uncertainty reward is simple but somewhat novel.

[1] Mei, Z., Zhang, C., Yin, T., Lidard, J., Shorinwa, O. and Majumdar, A., 2025. Reasoning about Uncertainty: Do Reasoning Models Know When They Don't Know?. arXiv preprint arXiv:2506.18183.

[2] Damani, M., Puri, I., Slocum, S., Shenfeld, I., Choshen, L., Kim, Y. and Andreas, J., 2025. Beyond binary rewards: Training lms to reason about their uncertainty. arXiv preprint arXiv:2507.16806.

---

> ### Author Rebuttal · Authors · 2026-03-30
>
> # Response to Reviewer NJEw
>
> We thank the reviewer for the constructive feedback and positive assessment. We address each concern below.
>
> ---
>
> ## Q1: On Calibration — How does PAQ improvement translate to calibration? Why not quantify uncertainties?
>
> **Calibration interpretation**: In our framework, the model's decisions are "answer" or "abstain." This is indeed a binary confidence scheme: answered questions are treated as 100% confident commitments, while abstentions represent 0% confidence. PAQ directly measures the actual accuracy of these high-confidence commitments, so an improved PAQ indicates that the model's definitive answers are more frequently correct, reflecting better alignment between expressed confidence and actual accuracy (i.e., improved calibration).
>
> **Why not quantify uncertainties**: We agree that uncertainty quantification (e.g., continuous confidence scores) offers distinct advantages for calibration, as calibrated uncertainty signals enable finer-grained decisions on whether to reveal answers. However, our design philosophy differs: we prioritize alignment with the general reasoning paradigm, where the model autonomously decides whether to express uncertainty during end-to-end reasoning, rather than relying on external thresholds. This choice is practically motivated by the observation that appropriate uncertainty thresholds vary significantly across tasks of different difficulty, and allowing the model to learn this boundary adaptively via RL circumvents the limitations of manual threshold tuning. We will include a discussion of these related works in the revision.
>
> ---
>
> ## Q2: Mixed Results Across Individual Datasets
>
> We provide deeper analysis:
>
> **(1) Training data distribution.** We use DAPO-Math-17k for training. Domain alignment varies across test sets—e.g., Minerva contains more interdisciplinary questions with a larger distribution gap, resulting in smaller gains. Standard GRPO shows similar patterns (almost no gain on Qwen3-8B), confirming this is a data distribution effect, not UCPO-specific.
>
> **(2) Limited PAQ/F1 gains on Qwen models.** Qwen3-8B has been extensively fine-tuned on large-scale math data, and its mathematical reasoning capability is near saturation. Achieving significant further gains on such a high baseline is inherently difficult, a phenomenon also reported in other RL works (e.g., [3]). As a result, UCPO's improvement on Qwen3-8B primarily stems from learning uncertainty expressions rather than enhancing reasoning itself, which precisely validates the core value of the UCPO framework.
>
> [3] Liu S Y, Dong X, Lu X, et al. Gdpo: Group reward-decoupled normalization policy optimization for multi-reward rl optimization[J]. arXiv:2601.05242, 2026.
>
> ---
>
> ## Q3: Accuracy Drop for General Tasks in Figure 5
>
> The accuracy in Figure 5 uses all questions as the denominator. In general tasks (multiple-choice), the model may have previously answered some low-confidence questions correctly by chance (e.g., 25% guessing probability for 4-choice). After UCPO training, these low-confidence but incidentally correct predictions become uncertainty outputs, causing overall accuracy to decline.
>
> However, PAQ (accuracy on answered questions) improves significantly: Qwen3-8B from 73.48% to 79.68%. UCPO effectively filters out lucky guesses, making definitive answers more reliable.
>
> ---
>
> ## Q4: Baseline Comparison with KnowRL
>
> The GRPO-UC variants ($r_u \in \lbrace 0.2, 0.5, 0.8 \rbrace$) serve as a unified abstraction of KnowRL and TruthRL, which share the same core mechanism: fixed intermediate-value rewards for uncertain predictions in GRPO. We will explicitly clarify this correspondence in the revised Baselines section.
>
> ---
>
> ## Q5: How Does Inference-Time Scaling Affect Uncertainty Awareness?
>
> We conducted preliminary experiments on Qwen3-8B General Tasks under different sampling counts $n$:
>
> | Method | Metric | n=1 | n=4 | n=8 | n=16 |
> |--------|--------|-----|-----|-----|------|
> | GRPO (vote@n) | PAQ | 74.27 | 77.70 | 76.83 | 76.32 |
> |               | F1  | 73.91 | 75.43 | 75.80 | 75.84 |
> | UCPO (vote@n) | PAQ | 81.12 | 82.45 | 83.25 | **84.05** |
> |               | F1  | 71.78 | 70.23 | 72.23 | 73.02 |
> | UCPO (any-uncertain@n) | PAQ | 81.12 | 82.45 | 87.76 | **90.86** |
> |                         | F1  | 71.78 | 70.23 | 70.78 | 68.31 |
>
> - **vote@n**: Majority voting over $n$ samples, including uncertainty predictions.
> - **any-uncertain@n**: Classified as uncertain if any of the $n$ samples expresses uncertainty.
>
> **Key findings**: (1) UCPO at $n=1$ (PAQ 81.12%) already surpasses GRPO at $n=16$ (76.32%), showing UCPO provides high reliability even without scaling. (2) GRPO's PAQ declines after $n=4$, suggesting overconfident errors interfere with majority voting. (3) UCPO's any-uncertain@n strategy reaches PAQ 90.86% at $n=16$, demonstrating strong complementarity between uncertainty awareness and inference-time scaling. We will explore deeper integration in future work.

---

> > ### Author Rebuttal · Reviewer_NJEw · 2026-04-03
> >
> > Thanks for your detailed response, I have no further questions.

---

### Decision · Program_Chairs · 2026-04-30

**Decision:**

Accept (regular)

**Comment:**

This paper proposes UCPO, a principled RL framework for uncertainty-aware LLM training that addresses instability and reward hacking in prior methods. The key ideas (TAD and DURA) are well-motivated and effectively resolve the ternary reward imbalance. All reviewers find the paper technically sound with clear presentation and meaningful empirical gains in reliability. Concerns were satisfactorily addressed in the rebuttal, with reviewers maintaining or strengthening the accept recommendations. The remaining issues are mostly minor and do not detract from the core contribution.